# Just-in-time: Gaze guidance in natural behavior

**Ashima Keshava**[1]*, **Farbod Nosrat Nezami**[1], **Henri Neumann**[2], **Krzysztof Izdebski**[2], **Thomas Schüler**[2], **Peter König**[1,3]

**1** Institute of Cognitive Science, University of Osnabrück, Osnabrück, Germany, **2** Halocline GmbH & Co. KG, Osnabrück, Germany, **3** University Medical Center, Hamburg-Eppendorf, Germany

* akeshava@uos.de

**Data Availability Statement:** The experimental data and analysis code can be found at https://osf.io/gq6aj/.

## Abstract

Natural eye movements have primarily been studied for over-learned activities such as tea-making, sandwich-making, and hand-washing, which have a fixed sequence of associated actions. These studies demonstrate a sequential activation of low-level cognitive schemas facilitating task completion. However, whether these action schemas are activated in the same pattern when a task is novel and a sequence of actions must be planned in the moment is unclear. Here, we recorded gaze and body movements in a naturalistic task to study action-oriented gaze behavior. In a virtual environment, subjects moved objects on a life-size shelf to achieve a given order. To compel cognitive planning, we added complexity to the sorting tasks. Fixations aligned with the action onset showed gaze as tightly coupled with the action sequence, and task complexity moderately affected the proportion of fixations on the task-relevant regions. Our analysis revealed that gaze fixations were allocated to action-relevant targets just in time. Planning behavior predominantly corresponded to a greater visual search for task-relevant objects before the action onset. The results support the idea that natural behavior relies on the frugal use of working memory, and humans refrain from encoding objects in the environment to plan long-term actions. Instead, they prefer just-in-time planning by searching for action-relevant items at the moment, directing their body and hand to it, monitoring the action until it is terminated, and moving on to the following action.

## Author summary

Eye movements in the natural environment have primarily been studied for over-learned and habitual everyday activities (tea-making, sandwich-making, hand-washing) with a fixed sequence of associated actions. These studies show eye movements correspond to a specific order of actions learned over time. In this study, we were interested in how humans plan and execute actions for tasks that do not have an inherent action sequence. To that end, we asked subjects to sort objects based on object features on a life-size shelf in a virtual environment as we recorded their eye and body movements. We investigated the general characteristics of gaze behavior while acting under natural conditions. Our

**Funding:** This project was supported by the German Federal Ministry of Education and Research to PK and TS, grant number 16SV8052. The funders had no role in study design, data collection and analysis, decision to publish, or preparation of the manuscript.

**Competing interests:** The authors declare that they have no conflict of interest.

paper provides a comprehensive approach to preprocess naturalistic gaze data in virtual reality. Furthermore, we provide a data-driven method of analyzing the different action-oriented functions of gaze. The results show that bereft of a predefined action sequence, humans prefer to plan only their immediate actions, where eye movements are used to search for the target object to immediately act on, then to guide the hand towards it and monitor the action until it is terminated. Such a simplistic approach ensures that humans choose sub-optimal behavior over planning under sustained cognitive load.

## Introduction

In a pragmatic turn in cognitive science, there has been a significant push to study cognition and cognitive processing during context-dependent interactions within the constraints of the local properties of an environment [1, 2]. This pragmatic shift places willed/voluntary actions and proactive control at the heart of cognitive processing, where behavior is natural and not cued externally. Moreover, Engel et al. [3] have proposed that cognition encompasses the body, and bodily action can be used to infer cognition. Furthermore, eye movements provide a unique window to understand cognitive control in natural, real-life situations [4]. This requires a mobile setup that allows a human subject to be recorded as they actively interact in a controlled but unconstrained environment [5]. In recent years, virtual reality (VR) and mobile sensing have offered great opportunities to create controlled, naturalistic environments. Here, subjects' eye and body movements can be reliably measured along with their interactions with the environment [6–8]. Studying eye movements in mobile subjects gives us a richer, veridical view of cognitive processing for willed actions.

Seminal studies have investigated eye movement behavior in natural environments with fully mobile participants. In the pioneering studies of Land et al. [9] and Hayhoe et al. [10], subjects performed everyday activities such as making tea and sandwiches, respectively. These studies required subjects to enact a sequence of actions that involved manipulating objects one at a time to achieve a goal. Both studies showed that nearly all gaze fixations were primarily allocated in a task-oriented manner, and a negligible number of fixations were found in task-irrelevant locations. These experiments in naturalistic settings have revealed several distinct functions of information sampling during routine everyday tasks.

Investigation of habitual tasks has revealed a systematic timing between visual fixations and object manipulation. Specifically, fixations are made to target objects about 600ms before manipulation. Notably, Ballard et al. [11] proposed a "just-in-time" strategy that universally applies to this relative timing of fixations and actions. Fixations that provide information for a particular action immediately precede that action and are crucial for the fast and economical execution of the task. Moreover, Land & Hayhoe [12] have broadly categorized fixations while performing habitual tasks into four functional groups. "Locating" fixations retrieve visual information. "Directing" fixations acquire the position information of an object, accompany a manipulation action, and facilitate reaching movements. "Guiding" fixations alternate between manipulated objects, e.g., fixations on the knife, bread, and butter while buttering the bread. "Checking" fixations monitor whether the task constraints have been met. These fixation categories have been corroborated by Pelz & Canosa [13] and Mennie et al. [14]. Hence, there is a just-in-time strategy while performing common tasks where gaze is specifically allocated toward searching for a target, guiding hands, and monitoring the action as it unfolds.

Based on the above observations, Land & Hayhoe [12] have proposed a framework that outlines the flow of visual and motor control during task execution Fig 1A. The process

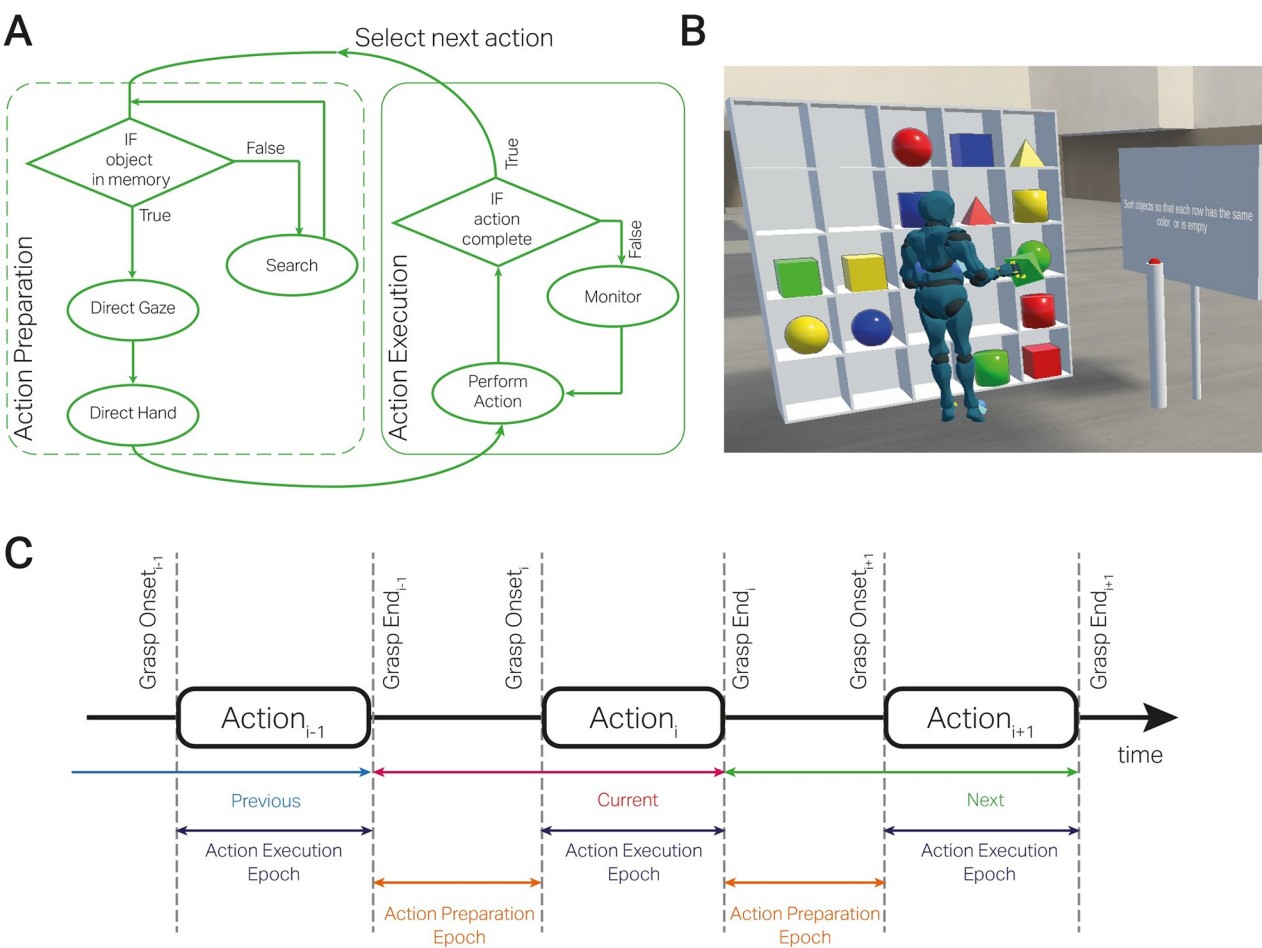

**Fig 1. Study design. A**: Schematic of motor and gaze control during performance of natural tasks. A cognitive schema selects between the object of interest, the corresponding action, and the monitoring of the action execution. If the object's location is not in memory, a visual search is undertaken to locate the object, the body and hands are directed towards it in preparation for the action execution, and finally, the action is monitored until the behavioral goal has been achieved. **B**: Experimental Task. In a virtual environment, participants sorted 16 objects based on two features, color or shape, while we measured their eye and body movements. The objects were randomly presented on a 5x5 shelf at the beginning of each trial. Participants were cued to sort objects by shape and color. Trials, where objects were sorted based on just one object feature (color or shape), were categorized as EASY trials. Conversely, the trials where sorting was based on both features (color and shape) were categorized as HARD trials. All participants performed 24 trials (16 easy and eight hard) without time constraints. **C**: Labelling of the continuous data stream into action preparation and execution epochs. To study the function of eye movements, we divided each trial into action preparation and execution epochs. The action execution epochs start from the grasp onset till the grasp ends for each object displacement. In contrast, the action preparation epochs start from the end of the previous grasp and the onset of the current object displacement.

summarizes the various operations that *must* occur during an object-related action', i.e., individual actions performed on separate objects to achieve the desired goal. Each schema "specifies the object to be dealt with, the action to be performed on it, and the monitoring required to establish that the action has been satisfactorily completed." [15]. In the action preparation period, the gaze control samples the information about the location and identity of the object and directs the hands to it. Subsequently, in the action execution period, the motor control system of the arms and hands implements the desired actions. Here, vision provides information about where to direct the body, which state to monitor, and determine when the action must be terminated. Thus, a 'script' of instructions is sequentially implemented where eye

movements earmark the task-relevant locations in the environment that demand specific attentional resources for that action.

An implicit theme in the studies investigating behavior in natural tasks (e.g., tea-making, sandwich-making, hand-washing) is that these tasks have an organized and well-known structure. They involve specific task-relevant objects, object-related actions such as picking up the teapot, pouring water, etc., and a predefined 'action script' for executing the tasks. Therefore, they study eye movements under strict control of a task sequence. Moreover, these tasks are over-learned as they are part of routine actions for a healthy human adult. For these habitual tasks, cognitive schemas allocate gaze for specific information retrieval (locating, directing, guiding, monitoring) and are likely not executed under deliberate conscious control. Hence, it is unclear how cognitive schemas control gaze where an internal task script is unknown.

Furthermore, a core feature of Land and Hayhoe's gaze and motor control framework is its sparse use of visual working memory. In this framework, low-level cognitive schemas are activated successively as needed. For example, it states that a visual search is performed if the object's location is not in memory. The model does not distinguish whether or how an additional cognitive load would affect the activation of each schema in its given sequence. As Hayhoe & Ballard [16] state, fixations provide information at different levels, such as for behavior, arbitration, and the broader context. At the level of individual behaviors, fixations provide the state of the environment and facilitate computations of whether a behavioral goal has been satisfied. A set of active behaviors might make perceptual and physical demands for attentional resources for which gaze can only be allocated sequentially. Hence, a mechanism is needed at the arbitration level to mediate conflicting resource demands. Finally, at the context level, gaze provides information to enable a set of context-specific behaviors, considering the state of the environment and the agent's goal. In well-learned tasks, action schemas are triggered automatically with little conscious control. However, when a task is relatively complex and requires top-down planning of subsequent actions, more information must be retained in memory as multiple schemas compete for attentional resources. Hence, a comprehensive action plan would need to maintain information about the object in working memory, how it should be manipulated, the conditions that signal the desired end state of the manipulation, and the subsequent actions specific to the context. As working memory is limited, it is pertinent to understand how attentional resources are arbitrated within the context of undefined action sequences and additional cognitive load.

Several computational models have been proposed to describe strategies of sequential gaze allocation. These strategies are based on how future rewards influence the subsequent fixations. The simplest strategy posits that the visual system selects the location where the task-relevant criterion (e.g., information about the search target or immediate reward) is maximal [17–19]. A more sophisticated strategy, proposed by other models, suggests selecting a target to maximize the search task criterion after the next gaze shift [20–22]. For example, the 'ideal-searcher' model will saccade between two potential targets to best decide the correct target. The above strategies correspond to a myopic or greedy observer model that selects each gaze target individually. Furthermore, a third strategy exists where sequences of gaze shifts are planned to maximize the overall task performance. In this regard, Hoppe & Rothkopf [23] have shown that human fixation behavior was better explained by the model of a planning observer that planned a sequence of saccades compared to a greedy observer, which selects only a single saccade at a time. Thus, there is evidence gaze sequences are planned optimally under uncertainty. However, it remains unclear whether optimal probabilistic planning to search for visual targets also corresponds to optimal decision-making in real-world settings.

The present study explored oculomotor behavior during novel tasks and the influence of cognitive load in the context of Land & Hayhoe's framework. In a new paradigm, we designed

novel tasks that did not have an associated predefined sequence of actions. Hence, the tasks required participants to plan actions for the specified context. Furthermore, we created two types of tasks that varied in cognitive load and required keeping multiple object features in memory. In virtual reality, we asked human volunteers to sort objects on a life-size shelf based on the object's features. The complexity of the task depended on sorting based on one or two object features. We simultaneously recorded gaze positions and body movements while subjects performed the tasks. Thus, our study investigates the mechanism of gaze allocation in novel tasks with varying complexity in a naturalistic environment.

Our study explored the characteristics of action-related gaze behavior for different task demands. At the outset, we tested the hypothesis of whether additional cognitive load influenced behavioral outcomes. In this regard, we were interested in the duration of preparation and execution of actions and the optimality of the action choices. Hence, optimal planning would indicate an optimal use of working memory where an action sequence is devised in advance but requires additional time. Similarly, with optimal planning, the execution durations would not differ between task complexities. Conversely, if subjects do not formulate a high-level plan for the action sequences, the preparation and execution of these actions would differ considerably, and more complex tasks would show proportionally longer durations. We further explored the corresponding action-related gaze behavior and its differences concerning cognitive load. With optimal planning, we expected initial fixations to encode the objects and corresponding action plans into memory. If an object's location is kept in memory for a considerable time, the added task complexity would not markedly increase visual search at the level of preparing for and executing actions. However, when planning is sub-optimal, data would show lower search behavior and greater just-in-time fixations corresponding to choosing objects randomly and then figuring out how to move them, resulting in a greater visual search for target objects during the execution period. Finally, we explored the influence of task complexity on the latency of fixations on task-relevant regions. In this way, we tested Land's theoretical framework for allocating attentional resources in novel and complex tasks.

## Results

Fifty-five human volunteers performed object-sorting tasks in a virtual environment. They sorted objects on a 5x5 shelf that measured approximately 2m in width and height. The tasks were based on sorting 16 objects of four colors (red, blue, yellow, green) and shapes (cube, sphere, pyramid, cylinder). Fig 1B illustrates the experimental setup. We experimentally modulated the task complexity into EASY and HARD trials. In the EASY trials, we gave participants instructions such as "Sort objects so that each row has the same color or is empty," and in the HARD tasks, an example trial instructed them to "sort objects so that each row has each unique color and unique shape once." Participants were not given any other instructions on how to complete the tasks. Hence, their action choices were not cued to follow a set strategy. Participants wore an HTC Vive Pro Eye virtual reality head-mounted display (HMD) as they performed the task. Using the HMD, we measured their eye and head movements simultaneously, while the HTC hand controllers measured their hand movements and grasping behavior. The eye and body tracking sensors were calibrated regularly throughout the experiment session.

We utilized two data streams for this study: the cyclopean eye (average of the left and right eye) direction vector and hand movement position. At the outset, we rejected data from participants where the sensors malfunctioned mid-experiment. We derived the angular velocity of the cyclopean eye from the direction data. We then used a median absolute deviation-based saccade detection method to cluster gaze samples into either fixation or saccade. Following

this, we removed fixations lasting less than 100 ms. Further, we rejected the fixations with a duration greater than 3.5 median absolute deviation. This step rejected 2.4% of the outlying gaze data.

Next, we derived the onset and end of a grasping action from the VR hand controller's trigger button, i.e., the time when a participant pressed the button with their right index finger to pick up an object and the time point when they released the button after moving it to a desired location. We did not consider grasps where the pickup and drop-off locations were the same. Finally, trials with fewer than three grasps were rejected. Also, we rejected the trials with more than 1.5 times the inter-quartile range (IQR) of the number of grasps by all participants. This step helped remove trials where subjects had outlying action choices. Hence, after pre-processing, we were left with 12403 grasping actions over 999 trials from 48 participants.

As a final step, we segmented the data stream into the action preparation and execution epochs. As illustrated in Fig 1D, these epochs marked the end of the previous grasp to the start of the immediate grasp and the time from the beginning of the current grasp to its end, respectively. Hence, every grasping action had a preparation and an execution phase with concurrent gaze data. Further, using the gaze data stream, we defined regions of interest (ROI) on the target object that was grasped and the target shelf where it was moved. These ROIs were further classified into the previous action, the current action, and the following action. The regions that did not correspond to targets in the action sequence were categorized as 'other objects' and 'other shelves.' This step prepared the data to demonstrate a "flow" of fixations from one action-relevant region to another during both the preparation and the execution phase of the grasping action.

## Influence of task complexity on natural behavior

In this study, the primary object-related action was to iteratively prepare and execute the pickup and placement of objects until they were sorted according to the task instructions. We used four measures to account for the behavioral differences between the task complexities. First is trial duration, i.e., the total time to finish the tasks. Second is the action preparation duration, the time between dropping the previous object and picking up the current target object. Third is execution duration, i.e., the time to move the target object to the desired location on the shelf. Fourth, the number of object displacements or "moves" made by the participants to complete the EASY and HARD tasks. These measures characterize the behavioral outcomes influenced by the task complexity.

Fig 2A shows the trial durations in EASY and HARD tasks. A two-sample independent t-test showed that the duration of the two task types was significantly different ($t = -10.92$, $p < 0.001$) where EASY tasks were shorter (Mean = 51.15$s$, SD = ±10.21) compared to HARD tasks (Mean = 123.46 ± 44.70$s$). Hence, the increase in task complexity markedly increased the duration of the tasks by a factor of two.

At a more granular level, we were interested in how task complexity affected the average action preparation and execution duration. For EASY tasks, the mean preparation epoch duration was 2.10$s$ (SD = ±0.78), and the mean execution epoch duration was 1.76$s$ (SD = ±0.27). The mean preparation epoch duration for HARD trials was 4.05$s$ (SD = ±1.99), and the mean execution epoch duration was 2.15$s$ (SD = ±0.58). Fig 2B depicts the mean and variance of the preparation and execution duration for the EASY and the HARD tasks.

We tested the hypothesis that the complexity of HARD trials increased the action preparation and execution duration. We used a linear mixed effects model with epoch duration for a given grasp as the dependent variable and trial type (EASY, HARD) and epoch type (PREPARATION, EXECUTION) as independent variables. We also added the grasp index, i.e., an

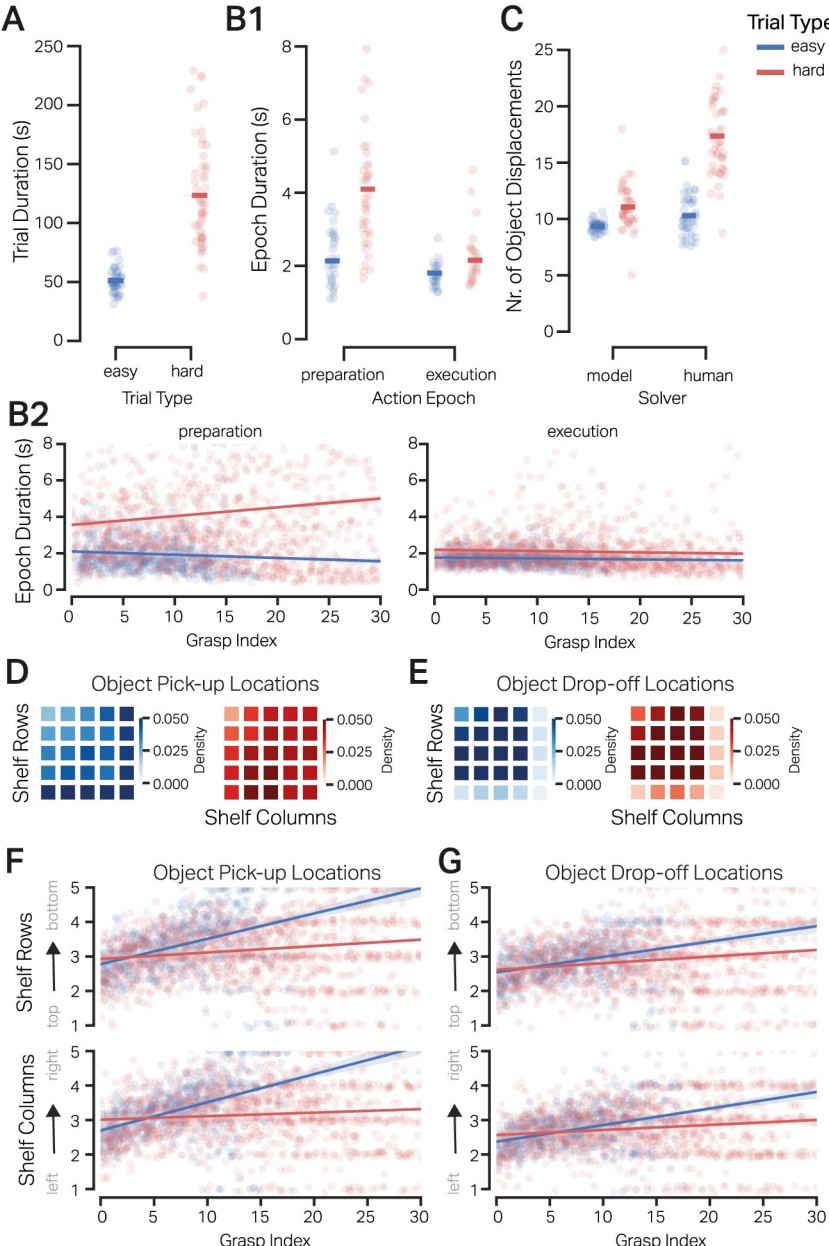

**Fig 2. Influence of task complexity on natural behavior. A**: Duration of the EASY and HARD tasks. Each dot represents the mean trial duration of each participant, and the solid bars indicate the mean duration over participants. **B1**: Duration of the action preparation and execution epochs under the EASY and HARD conditions. Each dot represents each participant's mean action epoch duration, and the solid bars indicate the mean epoch duration over participants. **B2**: Linear fit of epoch duration w.r.t. the index of the grasp/action in a trial (e.g., 0, 1, 2 indicates first, second, and third action of the trial respectively). Each point represents the average epoch duration of a subject for the corresponding grasp index. The plot shows that the preparation duration of the early grasps in HARD tasks was shorter than in the later grasps. This was not the case for the preparation or execution duration of EASY trials. **C**: Number of object displacements made by participants vs. those calculated by a greedy search model. The initial conditions of the task were inputted into a greedy model, yielding the optimal number of moves required to complete the respective trial. Each dot represents the mean object displacement of a participant and the model optimal counterpart over the EASY and HARD trials. The solid bars denote the mean over subjects. **D**: Propensity of picking up objects from a preferred location on the 5x5 shelf locations with blue heatmaps showing the probability density over EASY trials and red heatmaps showing density over HARD trials. The probability density shows that subjects tend to pick up objects from the rightmost column and bottom column for EASY trials (left). Conversely, in the HARD trials (right), subjects pick up objects from central locations. **E**: Propensity of dropping off objects to a preferred location on

the 5x5 shelf locations. The probability density shows subjects' systematic preference to move objects to locations other than the bottom row or rightmost column. **F**: Linear fit over average pickup row (top panel) and column (bottom panel) w.r.t grasp index. Each point represents a subject's average object pickup location for the corresponding grasp index. **G**: Linear fit over average drop-off row (top panel) and column (bottom panel) w.r.t grasp index. Each point represents a subject's average object drop-off location for the corresponding grasp index. The data show that participants predominantly manipulated and displaced objects from the top and left sides of the shelf at the beginning of a trial and progressed to the bottom and right parts later.

integer denoting the order of the object displacement in the trial (for example, 1st,2nd,3rd, etc.) as a covariate. The independent variables are effect-coded so that the regression coefficients can be directly interpreted as main effects. Table 1 summarizes the model results. Epoch duration was significantly different for the two trial types, where epochs of HARD trials were on average 1.31*s* longer than EASY trials. Irrespective of the trial type, the action preparation epochs were significantly longer than the execution epochs by 1.08*s*. An insignificant positive correlation of 0.002 with the grasp index indicates that overall, the epoch duration of initial moves did not differ from the latter ones. The preparation duration was significantly longer in the HARD trials, as shown by the interaction between the trial and epoch types. Furthermore, the difference in the slope of epoch duration and grasp index of HARD and EASY trials was insignificant. Similarly, the difference in the slope of the duration with respect to the grasp index between preparation and execution epochs was also not significant. However, the three-way interaction between trial type, epoch type, and the grasp index was 0.01 and significant, showing that the preparation duration of the HARD trials was significantly shorter at the beginning of the trial and increased linearly as the trial progressed. To summarize, the preparation and execution phases of the action were distinctly different, and task complexity specifically affected the duration of preparing actions and less so the time taken to execute them. Moreover, the significant interaction terms of the model suggest that in HARD trials specifically, participants made fast moves at the beginning of the trials and spent more time preparing their actions in the later part.

To determine whether participants optimally planned and sorted the objects during the trials, we compared their number of object displacements with an optimal greedy search algorithm. We fed the initial configuration of the shelf, as shown to the participants in each trial, into a greedy algorithm that found the maximal scoring moves and hence determined the

**Table 1. Model estimates of differences in duration of preparation and execution epochs w.r.t task complexity.**

| | Coefficient | 95% CI | t | p-value |
|---|---|---|---|---|
| **Model:** $log(duration) \sim 1 + trial\_type * epoch\_type * grasp\_index$ $+ (1 + trial\_type + epoch\_type + grasp\_index\|Subject)$ | | | | |
| trial type (HARD—EASY) | 1.31 | [1.26, 1.37] | 12.36 | $< .001^{***}$ |
| epoch type (PREPARATION—EXECUTION) | 1.08 | [1.01, 1.16] | 2.28 | **0.02**\* |
| grasp index | 0.002 | [0.00, 0.00] | 1.42 | 0.16 |
| **Interactions** | | | | |
| trial type: epoch type | 1.36 | [1.31, 1.40] | 17.13 | $< .001^{***}$ |
| trial type: grasp index | -0.003 | [-0.01, 0.0] | -1.44 | 0.15 |
| epoch type: grasp index | 0.002 | [0.00, 0.01] | 1.01 | 0.31 |
| trial Type: epoch type: grasp index | 0.01 | [0.00, 0.02] | 2.24 | **0.006**\*\* |

The table summarizes the fixed effects coefficients of the linear mixed model. The coefficients and 95%CI have been back-transformed to the response scale (in seconds) and can be interpreted as main effects.

optimal number of moves needed to solve that task. Fig 2C shows the mean number of object displacements made by the subjects and the optimal number of moves computed by the model for both EASY and HARD trials. In the EASY trials, subjects made an average of 10.30($SD$ = ±1.80) moves while the model optimal was 9.34($SD$ = ±0.45). In the HARD trials, participants made an average of 17.36($SD$ = ±4.04) moves when the optimal number was 11.06($SD$ = ±1.93). To test whether participants were optimal in their action choices, we fitted the data using a linear mixed model with the number of object displacements as the dependent variable and trial type (EASY, HARD) and solver (HUMAN, MODEL) as the independent variables. Again, as the independent variables were effect-coded, the regression coefficients can be directly interpreted as main effects. Table 2 details that the trial type significantly affected the number of object displacements, irrespective of the solver, where HARD trials demand an estimated 4.43(95%$CI$ = [3.81, 5.05]) more moves than the EASY trials. Similarly, irrespective of the trial type, participants made 3.32(95%$CI$ = [2.82, 3.90]) more object displacements than the optimal. Finally, there was a significant interaction between the trial type and solver, i.e., the participants made many more moves than the model, specifically in the HARD tasks. In sum, with increased task complexity, participants were more sub-optimal in the HARD trials than in the EASY condition.

As participants deviated from the optimal, especially in HARD trials, we checked if they employed any *salient* heuristics to accomplish the tasks. To do this, we used the measure of propensity to pick up and drop off objects from and to preferred locations on the shelf. We calculated the number of object displacements for each pickup and drop-off location. We then normalized the counts with the total number of displacements in the EASY and HARD tasks. Fig 2D shows that subjects preferred to pick up objects from the right-most column and bottom-most row of the shelf. Fig 2E shows that subjects had a propensity to drop the objects, leaving out the right column and bottom row of the shelf for both EASY and HARD trials. Given the sorting tasks where each trial began with a random initial configuration of objects on the shelf, we did not expect any systematic spatial biases at play. Further, the expectation was that subjects would move the objects randomly and not display a preference for object pickup and drop-off locations. These results show that subjects in our study systematically employed an arbitrary and non-task-specific heuristic to tackle the problem.

Next, we examined the object pickup and drop-off locations w.r.t. the trial progression. Fig 2F shows the average object pickup row and column for a given grasp index per subject. We calculated the Spearman correlation coefficient between the average pickup row and column and the grasp index. In EASY trials, the average pickup row and grasp index were highly correlated ($\rho$ = 0.49, $p$ < 0.001). Similarly, the average pickup column and the grasp index were highly correlated ($\rho$ = 0.54, $p$ < 0.001). In the HARD trials, the average pickup row and column were less correlated with the grasp index at ($\rho$ = 0.21, $p$ < 0.001) and ($\rho$ = 0.14, $p$ < 0/001),

**Table 2. Model estimates of differences between model and human object displacements dependent on the task complexity.**

| | Model:<br>$ObjectDisplacements \sim 1 + trial\_type * solver\_type$<br>$+ (1 + trial\_type + solver\_type | Subject)$ | | |
|---|---|---|---|
| | Estimate | 95% CI | t | p-value |
| trial type (HARD—EASY) | 4.43 | [3.81, 5.05] | 14.01 | < **.001***** |
| solver type (HUMAN—MODEL) | 3.36 | [2.82, 3.90] | 12.16 | < **.001***** |
| trial type: solver type | 4.90 | [4.25, 5.55] | 14.78 | < **.001***** |

The table summarizes the fixed effects coefficients of the linear mixed model.

respectively. Thus, subjects manipulated objects from the upper rows and leftward columns in the early moves and progressed to the lower and rightward parts of the shelf in later moves.

We found a similar behavior during object drop-off. Fig 2G shows the average object drop-off row and column for a given grasp index per subject. In EASY trials, the average drop-off row and column were positively correlated with the grasp index at ($\rho = 0.37$, $p < 0.001$) and ($\rho = 0.40$, $p < 0.001$). In HARD trials, the average drop-off row and column were less positively correlated with the grasp index at ($\rho = 0.22$, $p < 0.001$) and ($\rho = 0.20$, $p < 0.001$). Taken together, subjects initially manipulated objects from the top-left parts of the shelf and progressed to the lower-right locations towards the end of the trial.

Manipulating objects from left to right and top to bottom as a strategy was much more salient in the EASY trials than in the HARD trials. Compared to preparation duration in EASY trials, we can conclude that this strategy facilitated low-effort planning, as objects were simply displaced from left to right and top to bottom. Also, in the HARD trials, this strategy would have required less planning at the start of a trial as objects were moved quickly, but additional cognitive effort as the trial progressed and conflicts from object placements arose, leading to longer planning durations. This sub-optimal spatial heuristic would explain the longer preparation time in the HARD trials and the higher number of moves. To conclude, subjects preferred to offload cognitive effort on the environment by adopting sub-optimal and non-task-specific spatial heuristics.

Finally, we ascertained whether leaving out the bottom row and right column alone would explain participants' sub-optimal number of moves. We constrained the greedy search model by penalizing any object displacement to the last (bottom-most) row or the last (right-most) column. The resulting model performance did not differ significantly from the unconstrained model with 9.99 ± 0.4 moves in the EASY and 11.99 ± 1.22 moves in the HARD tasks. However, the constrained model could not return a solution for 17.5% of the HARD tasks overall. This is likely because the model uses a greedy approach that disregards less-scoring solutions at every step. Thus, a solution could have been found later in the decision tree by traversing other less-scoring moves. Hence, participants' leaving out the last row and column alone does not explain the large number of moves. Rather, the sequential manipulation of objects from left to right and top to bottom, as observed in the data, explains the sub-optimal outcome better.

The above behavioral results show that the categories of action epoch (preparation, execution) and task complexity (EASY, HARD) have distinct differences and interactions. Task complexity markedly affected the time taken to plan each action, with little influence on the execution duration. With increased task complexity, participants were increasingly sub-optimal in their actions. Lastly, although participants spent more time planning their immediate moves, especially within HARD tasks, they did not plan optimally and instead relied on a sub-optimal spatial heuristic.

## Action-locked gaze control

In the previous section, our analysis showed that task complexity significantly influenced behavior in the preparation phase and less so in the execution phase. Next, we examined the role of gaze control in the preparation and execution of these actions and how it is affected in complex task conditions. We also undertook an analysis that would generalize over activities that require planning successive manual actions from one object to the next. We labeled eight regions of interest (ROIs) for each object displacement or move. These ROIs denote the object and shelf locations handled in the previous, current, and next object displacements. Specifically, the current target object represented the object currently being displaced, and the current

target shelf was the location where the object would be placed. Similarly, the previous target object and previous target shelf represented the object handled in the previous object displacement and the shelf location where it was placed. It should be noted that the previous target object and the target shelf are spatially the same, as displacing the object has already been accomplished. The next target object and target shelf represented the objects and locations used after the current object displacement ended. Lastly, the 'other objects' and 'other shelves' categories represent objects and shelf locations that did not correspond to the sequence of previous, current, and subsequent actions. Together, these ROIs helped us investigate the flow of gaze control from one action to the next during the two phases of each action.

First, the analysis focused on the absolute timing of fixations on the eight ROIs time-locked to the grasp onset, i.e., when the hand makes contact with the current target object and the VR hand controller trigger button is pressed. For the analysis, we chose the period from 3s before action onset to 2s after. The time period selected approximately corresponds to the average duration of the preparation and execution epochs over the two trial types. We were also interested in the influence of task complexity on the average fixation behavior. Fig 3A shows the time course of the proportion of fixations on the eight ROIs for the EASY and HARD tasks.

In the action preparation phase, the average fixation profile shows essential aspects of oculomotor control 3s before the grasp onset. At the beginning of the preparation phase, there is an equal proportion of about 30% of fixations on the previous target object (dark blue trace), 'other objects' (dark grey trace), and 'other shelves' regions (light grey trace). The comparable proportion of fixations on previous targets and the 'other' categories indicates a visual search for task-relevant objects/shelves at the start of the preparation period. Subsequently, as the fixations on the previous targets and 'other objects' decrease, there is a steep increase in the proportion of fixations on the current target object (dark red trace), reaching a maximum of 68% of fixations at the point of grasp onset (time = 0s). Throughout the preparation period, the proportion of fixations on the next target object (dark green trace) and shelf (light green trace) remains comparatively small. The large proportion of fixations in the action preparation phase devoted to the current target object shortly before the hand makes contact with the object indicates just-in-time fixations used to direct and guide the hand toward the target object.

In the action execution phase, up to 2s after the grasp onset, the proportion of fixations on the different ROIs reveals further action-relevant oculomotor behaviors. Firstly, with time, the proportion decreases from the current target object and increases over the current target shelf (light red trace), where approximately 40% of the fixations are allocated around 700ms after grasp onset and well before the end of the action execution at 1.91s ($95\%CI[1.81, 2.01]$). Secondly, before the end of the action execution, there is an increase in the proportion of fixations on the next target object. The presence of approximately 10% of fixations on the next target object before the end of the current action execution shows that the target for the next action is located shortly before ending the current action in a number of trials. Moreover, about 20% of the fixations are devoted to the 'other' ROIs after 1s. This is again indicative of a search process for task-relevant objects/shelves. In a nutshell, during the action execution phase, the majority of fixations suggest two primary functions of gaze: first, to aid the completion of the current action by fixating on the shelf location in anticipation of the object drop-off, and second, to commence an object search for the next action when the target is not already decided.

Importantly, the proportion of fixations on the eight ROIs in the EASY and HARD tasks does not deviate dramatically across time. Even though there are large behavioral differences in the EASY and HARD tasks, both in terms of time taken to prepare for the actions and the number of actions required, the fixation profiles time-locked to the grasp onset are comparable. Most importantly, the fixations are made just in time, where gaze shifts occur systematically in sync with the previous, current, and next action sequences. Hence, we can conclude

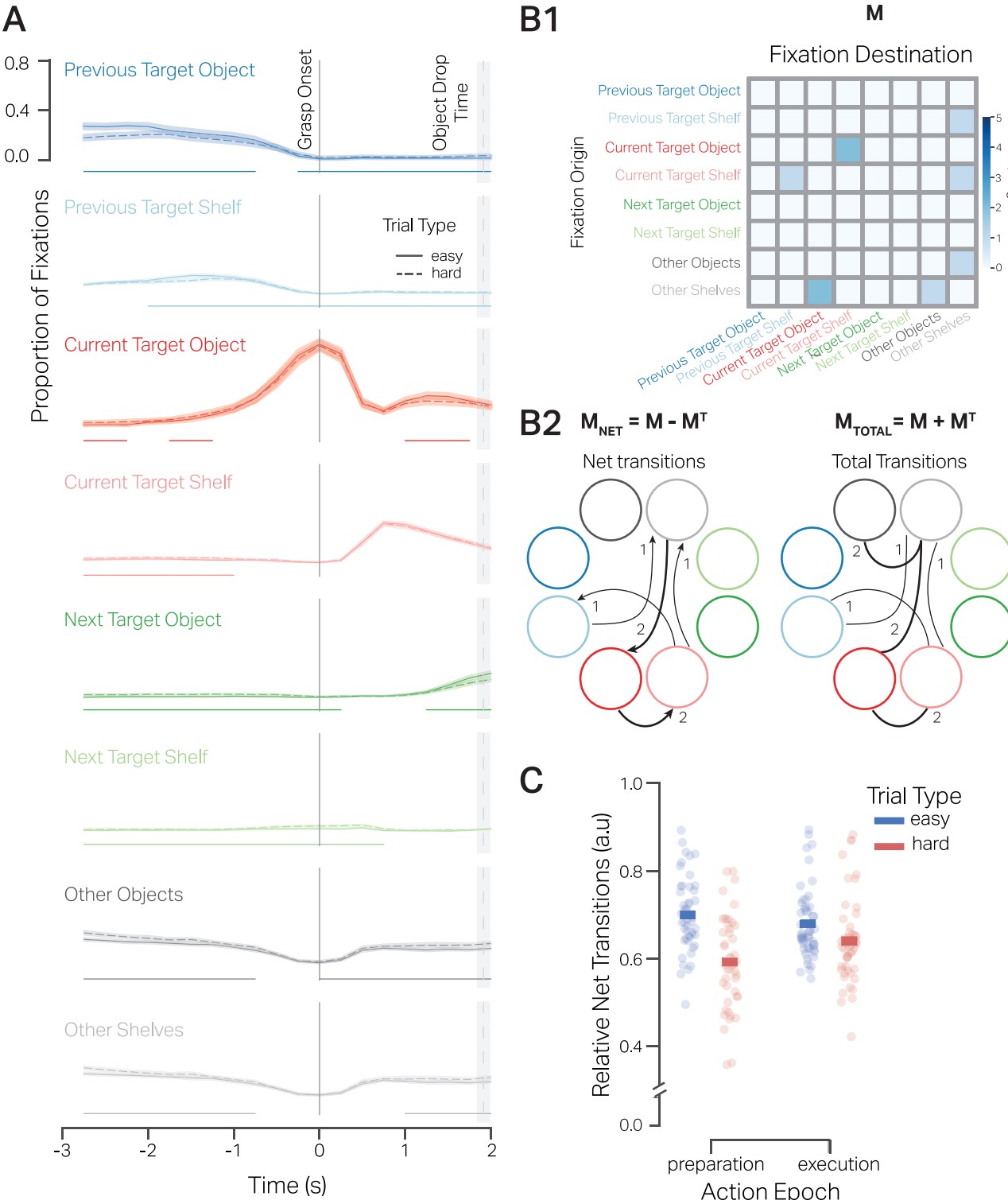

**Fig 3. Object fixations aligned to action onset. A**: Proportion of fixations on the eight regions of interest time-locked to grasp onset (time = 0s) from 3s before the onset to 2s after in the EASY tasks(solid trace) and HARD tasks(dashed trace). In each time bin of 0.25s, the proportion of fixations on all 8 ROIs for a trial type adds to 1. The shaded regions show the 95% CI of the mean proportion of fixations at each time-bin across all subjects. The dashed vertical line and shaded region denote the mean (time = 1.99s)and 95%CI [1.81s, 2.01s] of the end of the action execution. The horizontal traces (corresponding to the same color of an ROI) at the bottom of the line plots correspond to the periods where the proportion of fixations on a given ROI was significantly different between the EASY and HARD trials as computed by a cluster permutation test. **B1**: Exemplar transition matrix denoted as **M**

for gaze switching between ROIs in a preparation epoch. **B2**: The net ($M_{net}$) and total transitions ($M_{total}$) are calculated from matrix M. **C**: Relative net transitions (ratio of net transitions and total transitions) for action preparation and execution (abscissa) in the two trial types (EASY and HARD). The bars denote the mean relative net transition across subjects, and the circles correspond to each participant.

that the average oculomotor behavior described above is driven just-in-time by overt actions and less by the additional cognitive load of the tasks.

There are, however, slight differences in the proportion of fixations due to the task complexity. These differences point to aspects of the oculomotor behavior subject to the cognitive load imposed by task complexity. We used a cluster permutation analysis on the average fixation time course over each ROI to quantify these differences. The analysis revealed periods where the proportion of fixations significantly differed in the two task types on each of the eight ROIs. S1 Table details the start and end of the periods where the proportion of fixation on the ROIs significantly differed between the EASY and HARD trials. The table supplements the horizontal traces at the bottom of the average fixations in Fig 3 and depict the start and end of the period when the differences in a given ROI were statistically significant.

An interesting characteristic of the task-specific periods of differences is that they occur early in the action preparation epoch. During HARD tasks, there are fewer fixations on the previous target object (dashed dark blue trace) and previous target shelf (dashed light blue trace) and more fixations on the other target objects (dashed dark grey trace) and shelf (dashed light grey trace). In HARD tasks, there are slightly more fixations on the next target object (dark green trace), shelf (light green trace), and on the current target object (dashed dark red trace) and target shelf (dashed light red trace). These early fixations indicate that task complexity increased the incidence of 'search' and 'locate' fixations. As the average action preparation epoch was longer in the the HARD trials, the analysis exemplifies the prolonged visual search undertaken due to task complexity.

Similarly, concurrent with the action execution, there are more fixations on the previous target object and shelf, the next target shelf, and the other objects in the HARD tasks. More fixations on the previous target object and shelf are characteristic of 'checking' fixations used to monitor the current task execution with respect to the task condition. Further, more fixations on the 'other' objects indicate a greater search for the next target object in the HARD tasks. With an increase in fixations on the other object ROIs in HARD trials, later in the execution phase, there are correspondingly fewer fixations on the current and next target object. As a proportional decrease is not seen in the fixations on the current target shelf, we can infer that subjects began search for the next target object before the immediate action was terminated. Thus, specifically in the execution epochs of HARD tasks, the average fixations on the different ROIs show the functions of task monitoring and searching for the next action targets.

To summarize, the results in this section support two broad findings. First, despite significant behavioral differences between the two task types, the eye movements show slight deviations. This supports the idea that eye movements are tightly coupled to the actions and less so with the cognitive effort added by task complexity. Secondly, the observed significant differences between EASY and HARD trials are compatible with a slightly increased focus away from the previous action towards the current and the next step in the action sequence. This includes an earlier reduction of fixations on previous targets to facilitate search, the requirement for additional checking fixations after action onset, and a later attendance towards the next target object. Thus, cognitive processing maintains a structured sequence of activations corresponding to searching and locating a target object, guiding hands to it, and monitoring the task.

## Gaze transition behavior

To further elucidate the concurrent cognitive processes, we focused on sequences of attended regions of interest in individual trials. Specifically, we computed transition matrices to capture fixations to and from each of the eight ROIs. Fig 3B1 shows the exemplar transition matrix for a preparation epoch in an EASY trial. Fig 3B2 shows the derived net ($M_{net}$) and total ($M_{total}$) transition matrices from the example transition matrix ($M$). Total transitions represent the total number of fixations to and from one ROI. On the other hand, net transitions denote the asymmetry in gaze allocation from one ROI to another. For example, an equal number of fixations to and from pairs of ROIs would result in zero net transitions between those ROIs. Similarly, greater transitions from one ROI to another than vice versa would result in greater net transitions. The net transitions emphasize a circularity in the fixation behavior where subjects would exhibit more return fixations to already looked-at objects. It would show subjects' scant use of working memory by not encoding already fixated-on objects into memory. In contrast, purposeful use of working memory would require fewer refixations, more forward fixations, and greater net transitions.

We used the metric mean relative net transitions as the ratio of net transition by total transition for both action epochs of a trial. Hence, we computed the mean relative net transition for the preparation and execution epochs per trial. With maximum relative net transitions equal to 1, we can expect an equal number of the net and total transitions between ROIs, indicating forward saccades between ROIs and showing strictly action-oriented gaze guidance to ROIs, e.g., from previous to current in the action preparation phase. Such a gaze behavior would show no intermediate refixations on the ROIs. Conversely, the minimum relative net transition of 0 would indicate an equal number of saccades from and to ROI pairs. Searching for target objects by fixating back and forth over the ROIs or switching gaze between ROI pairs to monitor the task progression would lead to 0 net transitions. Hence, relative net transitions would help distinguish the function of saccades between ROI pairs as locating or searching target objects or monitoring the task.

We tested the hypothesis of whether task complexity influenced the net transitions of fixations during action preparation and execution. From the previous analysis, we inferred search behavior from the proportion of fixations during the action preparation and execution phase. We hypothesized that in HARD tasks, subjects would show lower net transitions during the action preparation epochs due to increased search behavior and even lower net transitions during the execution epoch due to increased monitoring and searching for the next action targets.

The data shows subtle differences in the relative net transitions for the trial types and the action epochs Fig 3C. In the EASY trials, the mean relative net transition for the preparation epochs was $0.69 SD = \pm0.10$ and for the execution epochs $0.68 \pm 0.08$. In the HARD trials, the mean relative net transition for the preparation epochs was $0.59 \pm 0.11$, whereas it was $0.64 \pm 0.10$ in the execution epochs. We used a linear mixed effects model to fit the mean relative net transitions in a trial per action epoch and trial type. Table 3 details the model

**Table 3. Model estimates of differences in relative net transition w.r.t task type, epoch type.**

| Model:<br>$NetTransitions \sim 1 + trial\_type * epoch\_type + (1 + trial\_type * epoch\_type \mid Subject)$ | | | | |
|---|---|---|---|---|
| | Estimate | 95% CI | t | p-value |
| trial type (HARD—EASY) | -0.06 | [-0.09, -0.06] | -8.85 | **0.001***** |
| epoch type (PREPARATION—EXECUTION) | -0.02 | [-0.04, 0.01] | -1.76 | 0.08 |
| trial type: epoch type | -0.06 | [-0.08, -0.03] | -4.58 | $< .001$***** |

coefficients. A significant effect of trial type showed that HARD trials had lower relative net transitions than EASY trials. No effect of epoch type showed that overall preparation epochs did not differ from execution epochs regarding relative net transitions. However, there was a significant interaction between trial type and epoch type, showing that preparation epochs in HARD trials had significantly lower relative net transitions. Thus, increased task complexity was associated with decreased relative net transitions. Notably, with increased task complexity, the reduction in net transitions was greater in the action preparation epochs and less in the execution epochs.

The analysis above lends further evidence to differential gaze guidance due to task complexity, specifically in the action preparation epochs. The higher relative net transitions in the EASY trials suggest saccades were made towards the immediate action-relevant objects with little requirement for search. The lower net transitions in the HARD tasks suggest significantly higher gaze switching to and from ROIs in search of task-relevant objects, especially during action preparation. The influence of task complexity was not likewise pronounced in the action execution epochs, indicating eye movements were engaged in a similar function of monitoring the task progression, resulting in back-and-forth saccades between the previous, current, and next target ROIs in the action sequence. Hence, with increased task complexity, there is an increase in visual search for action-relevant items during action preparation and a negligible influence during action execution.

## Latency of fixations on task-relevant ROIs

To further study the temporal aspects of gaze guidance behavior during action preparation and execution, we were interested in the latency of the first fixations to the eight ROIs. As a measure of latency, we used the median time to first fixate on an ROI in each action epoch per trial. We hypothesized that if fixations are made to the task-relevant regions per a set cognitive scheme, we would see a strict temporal window where these fixations occur in the preparation and execution epochs. A cognitive schema that involves first searching for the targets, directing the hand towards the target, locating the next target, etc., would lead to first fixations on the 'other objects', then on the current target object, and then onto the next target object. On the other hand, if fixations do not occur based on a set cognitive schema behind it, the occurrence of fixations would not follow a temporal structure. Furthermore, to make the latency comparable across the different durations of preparation and execution epochs, we normalized it by the duration of the given epoch. Hence, the normalized latency of the first fixation to the eight ROI provides a temporal measure of when a schema is initiated during the preparation and execution of an action.

Here, we describe the latency of the first fixations on the eight ROIs normalized by the epoch duration. Fig 4A shows the distributions of the first fixations on the eight ROIs in the preparation phase. The distributions show a flow of fixations from the previous target object to other objects and shelves and then to the current target objects. Importantly, the first fixations on the current target object were at $0.42(SD = \pm0.13)$ in EASY trials and at $0.46(SD = \pm0.11)$ in the HARD trials. As shown in Fig 4A, the distribution of the fixation latencies shows a structured sequence of attending to task-relevant ROIs. Notably, the first fixation on the current target object occurs only after 40% of the preparation epoch duration has elapsed.

We modeled latency of first fixations dependent on ROI type and trial type using linear mixed models. S2 Table details the model coefficients of the linear regression model that compared the latency of the first fixation on the current target object with the rest of the ROIs and their interactions with respect to the task complexity. The model shows the first fixations go from the previous target object and shelf to the other shelves and objects, onto the next targets,

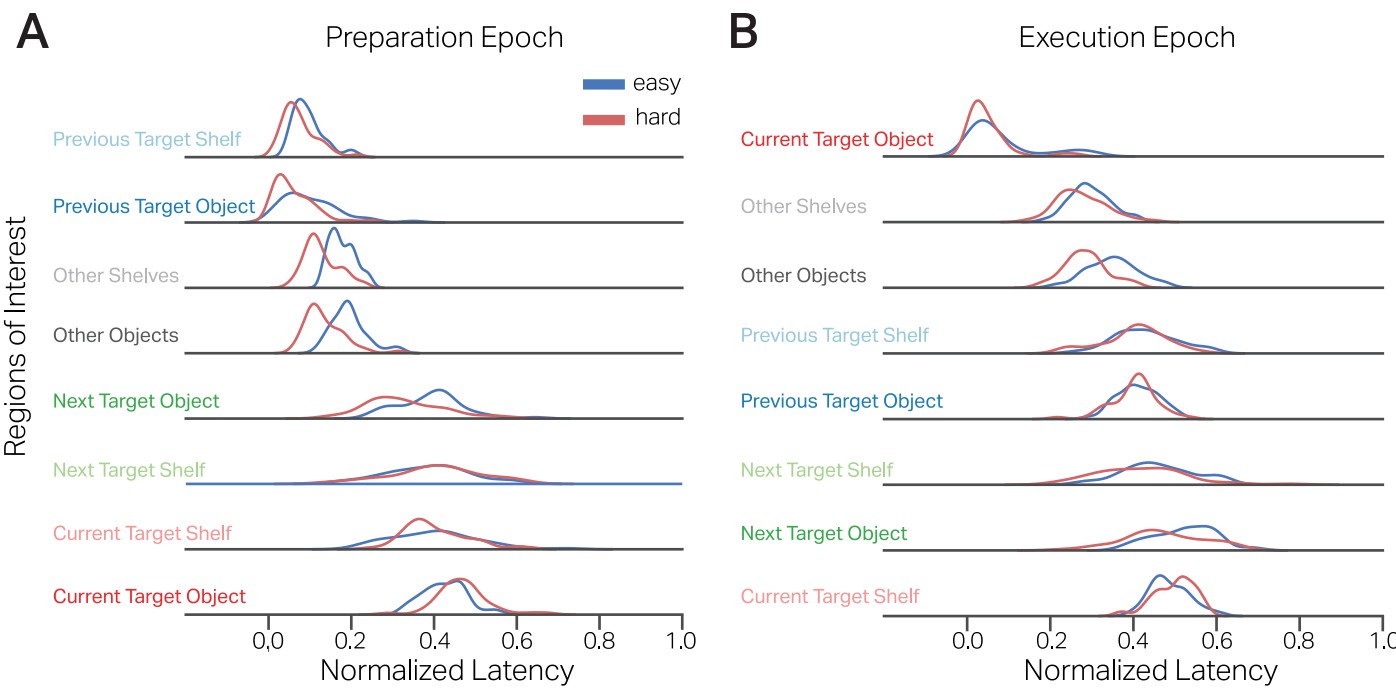

**Fig 4. Latency of gaze targets during action preparation and execution. A, B**: Distributions of median time to first fixation on the 8 ROIs for the action preparation and execution and trial types. The latency (abscissa) is normalized by the length of the preparation and execution epoch, respectively. The ordinate is sorted in ascending order of mean latency from epoch start.

and finally to the current target object. Furthermore, task complexity significantly affected the latency of first fixations on the previous target object and shelf, as well as the other and next target object, compared to the current target object. In HARD tasks, the first fixations on previous targets and the 'other' ROIs occurred significantly earlier than in the EASY tasks. Taken together, irrespective of task complexity, the action preparation epochs show a systematic progression of fixations from one ROI to another. This structured temporal sequence of fixations with a defined temporal window shows the fixations on the action-relevant ROIs are not incidental and are part of a set cognitive schema. Moreover, given the task complexity, the temporal profiles of these locating/searching fixations can change and occur earlier. Most importantly, the first fixations on the current target object occur last in sequence after more than 40% of the epoch time has elapsed, lending strong evidence to the just-in-time manner in which gaze is allocated to the immediate action requirements.

We similarly analyzed the latency of first fixations on the ROIs during the action execution epochs. Fig 4B depicts the distribution of the latency of first fixations on the different ROIs during the action execution epoch. Here, we see a similar temporal structure of first fixations that go from 'other' objects and shelves to previous targets and have similar latencies on the next target object and the current target shelf. Here again, the fixations on the current target shelf occur after half of the epoch duration has elapsed.

As before, we used a linear mixed model to model the latency of the first fixations dependent on the ROI type and trial type. S3 Table shows the model coefficients of the linear model that compared the latency of the first fixation on the current target object with the rest of the ROIs and their interactions with respect to the task complexity in the action execution epoch. To summarize, irrespective of the task complexity, there is a systematic sequence of first fixations on the eight ROIs during action execution. The early fixations on the previous target

object and shelf indicate these fixations are related to monitoring/checking the current action execution with respect to the previously handled object. Similarly, later fixations on the next target object and shelf indicate fixations used for planning the next action even before the concurrent action has been completed. Importantly, the increased task complexity did not affect the latency of first fixations on the task-relevant ROIs during action execution. Here again, the first fixation on the current target shelf, i.e., where the target object is placed, is fixated towards the end after 50% of the time has elapsed, strongly indicating a just-in-time nature of the action-relevant fixations.

Taken together, the latency of first fixations on the ROIs revealed the activation of specific cognitive schemas and their relative time of occurrence in the action preparation and execution epochs. Task complexity affected the timing of first fixations during action preparation and not in the execution phase. Furthermore, first fixations on the current targets occur after almost half the epoch time has elapsed, showing a just-in-time manner of gaze allocation.

The overall results suggest that action sequences are predominantly decided in a just-in-time manner, and task complexity influences distinct aspects of action-oriented fixations. The average fixation profile over the ROIs showed that fixations are driven from previous to current and next actions. Task complexity affected the proportion of fixations early during the preparation period to facilitate the visual search for targets and during the execution period for increased action monitoring. The sequential gaze transitions between ROI pairs further demonstrated an increased visual search associated with task complexity in the action preparation period. In contrast, task complexity did not influence gaze guidance during the execution phase. Task complexity also affected the latency of the first fixations on the ROIs in the preparation epochs but not in the execution epochs. Importantly, the latency analysis emphasized the just-in-time nature of gaze guidance, where fixations are made to the immediate action targets after half the action epoch (both preparation and execution) has passed. Finally, increased action preparation durations were associated with increased visual search. However, the prolonged visual search was not associated with optimal action choices. Instead, participants chose arbitrary spatial heuristics. Thus, gaze behavior was primarily explained by low-level cognitive schemas of searching, locating, directing, and task monitoring in the context of immediate actions.

## Discussion

In this study, we examined the mechanisms of gaze control in tasks that are not over-learned and routine. Building on the work of Land et al. [9], where eye movements were studied while preparing tea, the tasks in our study are novel because they do not have an inherent action sequence associated with them. Our experimental setup provided a way to capture oculomotor behavior for tasks that did not have a strict action sequence. By studying the dispersion and timing of fixations on the previous, current, and next action-relevant regions, we show a structured sequence of fixations that may be classified based on previous work. Moreover, we add to the current body of research by providing a data-driven approach to studying locating, directing fixations, guiding fixations, and checking fixations. Our analysis and findings generalize the occurrence of these fixations explicitly to the action sequence and completely disregard object identity.

Land et al. [9] proposed various functions of eye movements, from locating, directing, guiding, and checking during a tea-making task. Our study also shows the general signature of these fixations time-locked to the action onset and how they are impacted by cognitive load. Importantly, our study shows the occurrence of visual search early in the action preparation phase with increased search due to increased task complexity. The directing and guiding

                                                  

fixations occurred just before the action onset and were unaffected by task complexity. The checking/monitoring fixations occurred concurrently with the action execution as fixations on the previous action targets. Considering the latency and the proportional change in fixations on the task-relevant regions, our study further corroborates the cognitive schemas that sequentially support the preparation and execution of manual action.

Land & Furneaux [24] additionally elaborated on the schemas that direct both eye movements and motor actions. They proposed a chaining mechanism by which one action leads to another, where the eyes supply constant information to the motor system via a buffer. In our analysis, the temporal structure of the fixations on the ROIs lends further evidence of such a cognitive buffer that facilitates moving from one action to another where eye movements predominantly search and locate the regions necessary for immediate action. Interestingly, the increased cognitive load did not affect this simple mechanism of cognitive arbitration. Instead, we see a pronounced reliance on explicit visual search instead of encoding objects in memory. This is exemplified by the gaze transition analysis, where complex task conditions result in lower net transitions, indicating more refixations and fewer forward fixations. Thus, we show that cognitive schemas are activated to inform the current action with scant use of the visual working memory.

A notable feature demonstrated in the study is the just-in-time nature of action-oriented fixations. Here, the fixations on the target object occurred shortly before the action onset. Determining the causal relationship between these fixations and manual action is difficult in this study. A question remains: are gaze and arm movements planned together, or does the hand move only after the gaze has acquired the target's location? We did not explicitly study hand movement trajectory or velocity relative to the first fixation on the target object. However, the timing of the fixations on the different regions of interest indicates that once the target is acquired halfway through the preparation period, the remaining time is allocated to guiding the hand and body toward the target object. This is evidenced by a dip in the proportion of fixations on non-target objects and a sharp increase in fixations on the target object (Fig 3). This observation points to a model of greedy observer/actor model that uses visual information and plans only the immediate actions. Triesch et al. [25] have emphasized the task-specific nature of eye movements where information from fixations is used for computations only just in time to solve the current goal. Our study shows that a just-in-time strategy of gaze guidance consists of a search, locate, and act routine, and cognitive planning is primarily associated with the immediate action in the sequence.

Even though participants in our study spent a lot of time planning for the subsequent action under complex conditions, they were strikingly sub-optimal compared to the model solutions. There is considerable psychological evidence that supports the idea that anticipated cognitive demand plays a significant role in behavioral decision-making. Further, all things being equal, humans tend to take actions that reduce the cost of physical and cognitive effort [26]. Studies in human performance in reasoning tasks and combinatorial optimization problems [27] have revealed that humans solve these tasks in a self-paced manner rather than being dependent on the size of the problem. Pizlo & Li [28] found that subjects do not implicitly search the problem space to plan the moves without executing them. Hence, longer task durations do not necessarily lead to shorter solution paths. Instead, they showed that humans break the problem down into component sub-tasks, giving rise to a low-complexity relationship between the problem size and time to solution. Further, they showed that subjects use simpler heuristics to decide the next action. This is in line with Gigerenzer & Goldstein [29] who posit that humans use simplifying strategies to gather and search information that results in sub-optimal behavior. In more complex tasks, we observed higher planning durations and greater object displacements, but the action execution duration was similar to less complex tasks. Thus,

participants in our study do not necessarily demonstrate a lack of planning. Instead, the results show they divided the overall task into sub-tasks and planned which object needed to be handled and where it should be moved.

We observed a clear and general spatial heuristic employed by the participants. They used a strategy of arbitrarily organizing the objects from left to right and top to bottom and predominantly leaving out the last row and column. Such a simplistic strategy might be an artifact of reading direction habit [30, 31] or asymmetric attentional biases [32]. Also, a case could be made that moving objects to the last row was especially associated with intrinsic motor costs. Burlingham et al. [33] recently studied a phenomenon of motor "laziness" in naturalistic gaze selection, where eye and head motor effort contribute toward gaze guidance behavior. In this regard, rolling the eye and head downward would result in a motor effort to scan the shelf's state. This motor cost might have been exaggerated due to the weight of the VR HMD. Furthermore, as subjects repeatedly visually searched the shelf for target objects, clustering the objects closer together and leaving out empty spaces to the right and the bottom of the shelf would have been a simplifying strategy to constrain the search space. Hence, we can speculate that the observed spatial heuristic combines minimizing motor costs and simplifying ways to scan the task space.

From the perspective of embodied cognition [5, 34–37], humans off-load cognitive work on the environment, and the environment is central to the cognitive system. In this framework, cognition is situated and serves to produce actions. Draschkow et al. [38] recently showed that humans do not rely on working memory even when searching for objects is physically effortful. Our behavioral results show that subjects use spatial heuristics to constrain the search space and complete the tasks. This is indicative of exploiting the external world to reduce the cognitive effort of encoding multiple objects in memory and relying on repeated visual search to accomplish the task goals. Further, Droll & Hayhoe [39] have suggested that a just-in-time strategy is also likely used to lower the cognitive cost of encoding objects in the world into the visual working memory. Thus, the prevalence of the just-in-time strategy observed in our study is evidence of using the environment as an external memory.

The present state of VR has opened new research directions in naturalistic human behavior [25, 40–43]. However, with the natural eye, head, and body movements, the experimental data can be plagued with noise due to the slippage of the head-mounted display [44] and other body sensors. We calibrated the eye tracker regularly after every three trials to reduce such errors. We also calibrated the body-tracking sensors and the hand controller at the beginning of a trial and then halfway through the experiment. Moreover, participants in our study used VR hand controllers to grasp and move objects that could have seemed unnatural. To help them get accustomed to the VR environment, we gave them practice trials at the beginning of the experiment. Itaguchi [45] showed that grasping movements in VR do not differ quantitatively from grasping kinematics in the real world. Consequently, our study can inform real-world decision-making paradigms, even with its perceived shortcomings.

## Conclusion

In conclusion, in the present study, we studied the nature of willed action choices and the corresponding oculomotor control under novel and complex task conditions. Our results generalize past work on action-oriented eye movements to tasks that are not routine or over-learned. Our approach explicitly investigates the flow of gaze associated with the action sequence. The study demonstrates that action choices are far from optimal under novel and complex task conditions, and humans heavily rely on heuristics to reduce their cognitive load. Action-oriented gaze primarily corresponds to low-level cognitive schemas such as searching and

locating objects of interest, directing and guiding the body and hands to the object, and monitoring the task progression. More importantly, we show that cognitive processing and information sampling are tightly coupled with overt behavior and engaged in a just-in-time manner. Furthermore, we show that these cognitive schemas do not necessarily support optimal actions but only the production and execution of immediate actions.

## Materials and methods

### Ethics statement

The Ethics Committee of the University of Osnabrück approved the study (Ethik-37/2019). Before each experimental session, subjects gave their informed consent in writing. They also filled out a questionnaire regarding their medical history to ascertain they did not suffer from any disorder/impairments that could affect them in the virtual environment. After obtaining their informed consent, we briefed them on the experimental setup and task.

### Participants

A total of 55 participants (39 females, mean age = 23.9 ± 4.6 years) were recruited from the University of Osnabrück and the University of Applied Sciences Osnabrück. Participants had normal or corrected-to-normal vision and no history of neurological or psychological impairments. They either received a monetary reward of €7.50 or one participation credit per hour.

### Apparatus & procedure

For the experiment, we used an HTC Vive Pro Eye head-mounted display (HMD)(110˚ field of view, 90Hz, resolution 1080 x 1200 px per eye) with a built-in Tobii eye-tracker with 120 Hz sampling frequency and $0.5˚ − 1.1˚$ calibration error. With their right hand, participants used an HTC Vive controller to manipulate the objects during the experiment. The HTC Vive Lighthouse tracking system provided positional and rotational tracking and was calibrated for a 4m x 4m space. We used the 5-point calibration function provided by the manufacturer to calibrate the gaze parameters. To ensure the calibration error was less than 1˚, we performed a 5-point validation after each calibration. Due to the study design, which allowed many natural body movements, the eye tracker was calibrated repeatedly during the experiment after every three trials. Furthermore, subjects were fitted with HTC Vive trackers on both ankles and elbows and one on the midriff. The body trackers were calibrated subsequently to give a reliable pose estimation using inverse kinematics of the subject in the virtual environment. We designed the experiment using the Unity3D version 2019.4.4f1 and SteamVR game engine and controlled the eye-tracking data recording using HTC VIVE Eye-Tracking SDK SRanipal (v1.1.0.1).

The experimental setup consisted of 16 objects placed on a shelf of a 5x5 grid. The objects were differentiated based on two features: color and shape. We used four high-contrast colors (red, blue, green, and yellow) and four 3D shapes (cube, sphere, pyramid, and cylinder). The objects had an average height of 20cm and a width of 20cm. The shelf was designed with a height and width of 2m with five rows and columns of equal height, width, and depth. Participants were presented with a display board on the right side of the shelf where the trial instructions were displayed. Subjects were also presented with a red buzzer to indicate the end of the task.

### Experimental task

Subjects performed two practice trials where they familiarized themselves with handling the VR controller and the experimental setup. In these practice trials, they could explore the virtual

environment and freely displace the objects. After the practice trials, subjects were asked to sort objects based on the object features. There were two types of trials: EASY and HARD. Subjects were not limited by time to complete the task. The participants were not explicitly told about the task category and were told to follow the instructions given to them at the beginning. They were not given any other instructions on how to complete the task. Each subject performed 24 trials in total. Each trial type (as listed below) was randomly presented twice throughout the experiment. The experimental setup is illustrated in Fig 1B. The EASY trial instructions were as follows:

1. Sort objects so that each row has the same shape or is empty

2. Sort objects so that each row has all unique shapes or is empty

3. Sort objects so that each row has the same color or is empty

4. Sort objects so that each row has all unique colors or is empty

5. Sort objects so that each column has the same shape or is empty

6. Sort objects so that each column has all unique shapes or is empty

7. Sort objects so that each column has the same color or is empty

8. Sort objects so that each column has all unique colors or is empty

   The HARD trial instructions were as follows:

1. Sort objects so that each row has all the unique colors and all the unique shapes once

2. Sort objects so that each column has all the unique colors and all the unique shapes once

3. Sort objects so that each row and column has each of the four colors once.

4. Sort objects so that each row and column has each of the four shapes once.

## Data pre-processing

**Gaze data.** As a first step, using the eye-in-head 3d gaze direction vector for the cyclopean eye, we calculated the gaze angles for the horizontal $\theta_h$ and vertical $\theta_v$ directions. All of the gaze data was sorted by the timestamps of the collected gaze samples. Each sample's 3d gaze direction vector is represented in $(x, y, z)$ coordinates as a unit vector that defines the direction of the gaze in VR world space coordinates. In our setup, the $x$ coordinate corresponds to the left-right direction, $y$ in the up-down direction, and $z$ in the forward-backward direction. The formula used for computing the gaze angles are as follows:

$$\theta_h = \frac{180}{\pi} * \arctan\frac{x}{z}$$

$$\theta_v = \frac{180}{\pi} * \arctan\frac{y}{z}$$

Next, we calculated the angular velocity of the eye in both the horizontal and vertical coordinates by taking the first difference of the gaze angles with respect to time using the formula

below:

$$\omega_h = \Delta\theta_h / \Delta t$$

$$\omega_v = \Delta\theta_v / \Delta t$$

Finally, we calculated the magnitude of the angular velocity ($\omega$) at every timestamp from the horizontal and vertical components using:

$$\omega = \sqrt{\omega_h^2 + \omega_v^2}$$

We used an adaptive threshold method for saccade detection described by [46] to filter the samples where gaze was relatively stable. We selected an initial saccade velocity threshold $\theta_0$ of $200°/s$. All eye movement samples with an angular velocity less than $\theta_0$ were used to compute a new threshold $\theta_1$ as three times the median absolute deviation of the filtered samples. $\theta_1$ was used as the new threshold to filter samples, and the above process was repeated until the difference between $\theta_n$ and $\theta_{n+1}$ was less than or equal to $1°/s$. This way, we arrived at the cluster of samples that belonged to fixations, and the rest were classified as saccades.

After this, we calculated the duration of the fixations and saccades. To handle minuscule fixations and saccades, we labeled all samples with saccade duration less than 30 ms bracketed by fixations also as a fixation and merged the fixations before and after this interval. Similarly, we labeled all fixation samples with a duration of less than 50 ms, interspersed between two saccades, also as a saccade. Following this, we recalculated the fixation and saccade durations. This procedure suitably handles the misclassification of single samples or pairs of samples into fixations or saccades. Finally, we rejected all fixations with a duration greater than 3.5 times the median absolute deviation of the population fixation duration and fixations that were less than 100 ms long.

**Grasp data.** Subjects used the trigger button of the HTC Vive controller to virtually grasp the objects on the shelf and displace them to other locations. In the data, the trigger was recorded as a boolean with a default value of FALSE and set to TRUE when a grasp was initiated and then reset when the trigger was released to end the grasp. Using the position of the controller in the world space, we determined the locations from the shelf where a grasp was initiated and ended. We also removed trials where the controller data showed implausible locations in the world space. These faulty data can be attributed to sensors losing tracking during the experiment. Next, we removed grasping periods where the beginning and final locations of the objects on the shelf were the same. We calculated the interquartile range (IQR) of the participants' number of object displacements for the two trial types (EASY and HARD). To remove the outlying object displacements in the trials, we removed the trials with 1.5 times the IQR of object displacements. We also removed those trials with fewer than three object displacements.

After pre-processing, we were left with data from 48 subjects with 999 trials in total and $20.8 \pm 1.8$ trials per subject. Furthermore, we had eye-tracking data corresponding to 12, 403 object displacements with $12.3 \pm 5.11$ object displacements per trial.

Using the grasp onset and end events, we divided the continuous data stream into action preparation and execution epochs. Specifically, the action preparation phase starts from the end of a previous grasp onset till the start of the immediate grasp onset. Similarly, the execution epoch started from the start of the current grasp onset to the end of that grasp. The division of the data into these epochs is depicted in Fig 1C.

We further organized the data to examine the spatial and temporal characteristics of eye movements during the action preparation and execution epochs. We labeled the fixation based

on its location on the different objects and shelf locations into eight regions of interest (ROIs). These included fixations of targets in the previous, current, and next grasping actions. Specifically, the previous target object refers to the object handled in the previous action, and the previous target shelf is the shelf where the previous target object was placed. Similarly, the current target object refers to the object picked up and placed on the target shelf in the current action, and the next target object and next target shelf are in the immediate following action in the sequence. All other fixated regions that did not conform to the above six ROIs were categorized as 'other objects' and 'other shelves'. In this format, we could parse the sequence of fixations on the eight ROIs that are relevant for preparing and executing the object-related actions.

## Data analysis

**Influence of task complexity on behavior.** We used four measures to examine the grasping behavior during the experiment. First is trial duration, i.e., the total time to finish the tasks. Here, we tested the hypothesis that task complexity affected the total duration of finishing the tasks using a two-sample independent t-test. Second is the action preparation duration, which is the time between dropping the previous object and picking up the current target object. Third is execution duration, i.e., the time to move the target object to the desired location on the shelf. We tested the hypothesis that the action epochs durations differed based on the action epoch type and the task complexity. Finally, we were interested in the number of object displacements made by the participants to complete the tasks. We tested the hypothesis that participants were optimal in their action choices in the EASY and HARD conditions. Taken together, these measures capture the influence of task complexity on overt behavior.

We modeled the relationship between the action epoch duration dependent on the action epoch type (PREPARATION, EXECUTION) and the trial type (EASY, HARD). As the durations differed from the start of the trial to the end, we added the grasp index as a covariate in the model. All within-subject effects were modeled with random intercepts and slopes grouped by subject. The dependent variable was log-transformed so that the model residuals were normally distributed. The categorical variables trial type and epoch type were effect-coded [47] so that the model coefficients could be interpreted as main effects. The model coefficients and the 95% confidence intervals were back-transformed to the response scale. The model fit was performed using restricted maximum likelihood (REML) estimation [48] using the lme4 package (v1.1–26) in R 3.6.1. We used the L-BFGS-B optimizer to find the best fit using 20000 iterations. Using the Satterthwaite method [49], we approximated degrees of freedom of the fixed effects. The full model in Wilkinson notation [50] is denoted as:

$$Log(duration) \sim 1 + trial\_type * epoch\_type * grasp\_index$$
$$+(1 + trial\_type + epoch\_type + grasp\_index|Subject)$$

**Comparison of human vs. model performance.** To ascertain whether the action choices of the subjects were optimal, we compared their performance to a greedy search model. The model was designed to maximize the score of each possible move. The model operations are summarized in Algorithm 1. For the EASY tasks, a move's score depended on the number of identical object features, e.g., color in each row or column, where the score of each shelf configuration was the sum of the scores of each row. In the HARD tasks, the score of a move was calculated by summing the number of unique object features per row or column. In this way, the greedy model found the optimal moves by maximizing the score of the possible next move.

**Algorithm 1** Greedy Model

```
1: Create empty stack S for each shelf state
2: PUSH initial shelf configuration to S
3: while S is not empty do
4:    POP latest state sᵢ in S
5:    current score = score of state sᵢ
6:    for each row in sᵢ do
7:      for each object in row do
8:        for each row in sᵢ that is not current row do
9:          for each empty space in row do
10:             s_next = object in empty space
11:             new score = score of s_next
12:             if new score > current score and move not in S then
13:               PUSH s_next to S
14:               update current score to new score
15:             end if
16:          end for
17:        end for
18:      end for
19:    end for
20: end while
21: Return length of S as number of moves to final state
```

We modeled the relationship between the number of moves made by subjects in each trial dependent on the trial type (EASY, HARD) and the solver type (MODEL, HUMAN). As before, all within-subject effects were modeled with random intercepts and slopes grouped by subject. The independent variables were also effect-coded. The full model in Wilkinson notation [50] is denoted as:

$$ObjectDisplacements \sim 1 + trial\_type * solver\_type$$
$$+ (1 + trial\_type + solver\_type | Subject)$$

**Action locked gaze control.**   We analyzed the average fixation behavior aligned with the action onset. For each grasp onset in a trial, we chose the period from 3s before grasp onset to 2s after. We divided this 5s period into bins of 0.25 seconds and calculated the number of fixations on the eight ROIs described above. For each time bin, we calculated the proportion of fixations on the ROIs per trial type (EASY, HARD). We used the cluster permutation method to find the periods of significant differences between EASY and HARD trials for a given ROI. Here, we use the t-statistic as a test statistic for each time-bin, where t is defined as:

$$t = \sqrt{N} * \frac{x}{\sigma}$$

where x is the mean difference between the trial types, $\sigma$ is the standard deviation of the mean, and N is the number of subjects. We used a threshold for t at 2.14, corresponding to the t-value at which the p-value is 0.05 in the t-distribution. We first found the time bins where the t-value was greater than the threshold. Then, we computed the sum of the t-values for these clustered time bins, which gave a single value representing the cluster's mass. Next, to assess the significance of the cluster, we permuted all the time bins across trials and subjects and computed the t-values and cluster mass for 1000 different permutations. This gave us the null distribution over which we compared the cluster mass derived from the real data. To account for the multiple independent comparisons for the eight ROIs, we considered the significant clusters to have a Bonferroni corrected p-value less than 0.006 (0.05/8). In the results, we

report the range of the significant time bins for the eight ROIs for the two trial types and the corresponding p-values.

**Gaze transition behavior.** To examine the gaze transitions within the action preparation and execution epochs we created transition matrices that show the origin and destination locations of the fixations on the eight ROIs. A series of fixations on the ROIs in a given action epoch were transformed into a 2D matrix ($M$) with eight rows as the origin of the fixation and eight columns as the destination. The diagonal of the matrix was set to zeros as we were not interested in re-fixations on the same ROI. Using matrix $M$, we calculated the net and total transitions from and to each ROI. For every transition matrix **M** per action preparation and execution epoch, the net ($M_{net}$), and total ($M_{total}$) transition are defined as follows:

$$M_{net} = M - M^T$$

$$M_{total} = M + M^T$$

If subjects make equal transitions between the ROI pairs in both directions, we can expect zero transitions in $M_{net}$. Conversely, with strong gaze guidance towards a particular ROI, we would expect more net transitions. Hence, using the net and total transitions per action epoch, we calculated the relative net transitions (R) as:

$$R = \frac{\sum M_{net}}{\sum M_{total}}$$

We then took the mean of the relative net transitions per trial to measure asymmetric gaze guidance behavior in that trial.

We modeled the linear relationship of the relative net transitions dependent on the trial type (EASY, HARD), epoch type (preparation, execution), and their interactions. All within-subject effects were modeled with random intercept and slopes grouped by subject. The categorical variables trial type and epoch type were effect-coded. The full model in Wilkinson notation [50] is defined as:

$$Relative_{Transitions} \sim 1 + trial\_type * epoch\_type$$
$$+(1 + trial\_type * epoch\_type | Subject)$$

**Latency of first fixations on task-relevant ROIs.** Finally, we also calculated the latency of the first fixation on the eight ROIs from the start of the preparation and execution epochs. We computed the median time to first fixate on an ROI per trial. As the action preparation and execution epochs varied in duration, we normalized the time points by dividing them by the duration of the epoch. Thus, the normalized elapsed time from the start of an epoch is comparable to all epochs across trials and subjects.

We modeled the linear relationship of the normalized median time to the first fixation dependent on the trial type (EASY, HARD) and the eight ROIs and their interactions. We computed two models for the action preparation and execution epochs. All within-subject effects were modeled with random intercepts and slopes grouped by subject. The categorical variables trial_type and ROI were effect coded as before, and the model coefficients could be interpreted as main effects. For both models, we chose the latency of the first fixation on the current target object as the reference factor so that the latency of the fixation of all other ROIs

could be compared to it. The full model in Wilkinson notation [50] is defined as:

$$Fixation_{time} \sim 1 + trial\_type * ROI$$
$$+(1 + trial\_type * ROI|Subject)$$

## Supporting information

**S1 Table. Significant differences in average fixation profile.** Significant differences in the proportion of fixations on the ROIs in the HARD and EASY trials as revealed by a cluster permutation test with a Bonferroni corrected significance threshold of 0.006.
(PDF)

**S2 Table. Model estimates of latency of first fixations during action preparation epoch.** Fixed effects coefficients of the linear mixed model detailing estimated latency of first fixations w.r.t the current target object during the action execution epochs.
(PDF)

**S3 Table. Model estimates of latency of first fixations during action preparation epoch.** Fixed effects coefficients of the linear mixed model detailing estimated latency of first fixations w.r.t the current target object during the action execution epochs.
(PDF)

## Author Contributions

**Conceptualization:** Ashima Keshava, Thomas Schüler, Peter König.

**Data curation:** Ashima Keshava.

**Formal analysis:** Ashima Keshava, Farbod Nosrat Nezami.

**Funding acquisition:** Thomas Schüler, Peter König.

**Investigation:** Ashima Keshava.

**Methodology:** Ashima Keshava.

**Project administration:** Krzysztof Izdebski, Thomas Schüler.

**Resources:** Thomas Schüler, Peter König.

**Software:** Ashima Keshava, Henri Neumann.

**Supervision:** Peter König.

**Validation:** Ashima Keshava.

**Visualization:** Ashima Keshava.

**Writing – original draft:** Ashima Keshava.

**Writing – review & editing:** Ashima Keshava, Thomas Schüler, Peter König.

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
