## [Decision Letter · Decision Letter 0]

25 Aug 2022

Dear Ms Keshava,

Thank you very much for submitting your manuscript "Low-level Action Schemas Support Gaze Guidance Behavior for Action Planning and Execution in Novel Tasks" for consideration at PLOS Computational Biology.

As with all papers reviewed by the journal, your manuscript was reviewed by members of the editorial board and by several independent reviewers. In light of the reviews (below this email), we would like to invite the resubmission of a significantly-revised version that takes into account the reviewers' comments.

We cannot make any decision about publication until we have seen the revised manuscript and your response to the reviewers' comments. Your revised manuscript is also likely to be sent to reviewers for further evaluation.

Sincerely,

Aldo A Faisal

Academic Editor

PLOS Computational Biology

Wolfgang Einhäuser

Section Editor

PLOS Computational Biology

Jason A. Papin

Editor-in-Chief

PLOS Computational Biology

Feilim Mac Gabhann

Editor-in-Chief

PLOS Computational Biology

Reviewer's Responses to Questions

**Comments to the Authors:**

Reviewer #1: I think this is an interesting study examining how task complexity influences gaze patterns. The task itself is very clever in that the manipulation clearly influences the complexity of the planning stage, but not necessarily of the execution. I feel confident that something interesting could be found in this dataset, but the current presentation makes it very difficult for me to see what. I do not understand why many of the analyses are done in the way they are, or even at all. This does not mean that it was wrong to do them, or to do them in this way, but I think the reasoning needs to be clarified. In combination with the fact that many of the analyses seem superfluous to me (because the outcome is trivial) the paper left me with the feeling that I did not learn anything. I think the authors should really consider what they want to demonstrate or explore with this study, and focus on that issue. Doing so will probably involve completely rewriting the paper. The main issues that I have with the current presentation are explained below (in random order).

I think the authors need to explain the task much earlier, not in a methods section at the end of the paper. Now they force the reader to look back and forth. It is fine to keep details that are not needed for understanding the results for the end of the paper, but it is currently impossible to understand the results without first looking at the various instructions that specify whether a task is simple or complex. This problem starts already in the Author Summary, where the sentence starting with “By only studying the sequence of fixations based on past, current and next actions” makes no sense at all without knowing what you mean by ‘only’ or by the three types of actions.

The introduction is extremely broad, and introduces lots of aspects, but then never really becomes concrete. This is probably partially responsible for the whole study then failing to really answer any question. The most concrete prediction is that “an abundance of look-ahead saccades would be evidence of elaborate cognitive planning” (line 95), but this is then never really examined empirically, while lots of other details are. To do so would require a more detailed study of what is now considered the ‘other’ category, especially before grasping. Now, the data in Figure 3 suggest that there is no such elaborate cognitive planning.

There are several claims that are not explained in the results section as well. For instance, on line 137: “Further, the expectation was that the subjects would move the objects randomly and not display a preference for object pickup and drop-off locations”. Why? Surely there are some more comfortable locations or movements. Moreover, they may want to move less far (as indeed turns out to be something the authors require for movements to be ‘optimal’). It is also claimed that the tendency to move objects up and to the left is not optimal, but it is not clear why. In many cases there are presumably several equivalent options. In such cases it may not be sub-optimal to select a certain one. Of course figure 2 suggests that participants wanted to order the items in a certain way in the top left, and perhaps even did so when that meant moving one (or even several) items more than would be necessary if they used different rows and columns. Apparently they did not consider leaving a central row empty. This could just be the way they interpreted the instruction, or it could feel easier. It can probably account for much, if not all, of the “sub-optimal” behavior. This is maybe interesting, but not really relevant for the question.

On line 145 again ROIs are introduced without the reader knowing what they are. A few sentences later the reader is referred to the methods section. That really disturbs the flow of reading the paper. If you need to know this to understand the results it needs to be explained. This can be very brief: the current object and where to put it, and the object and place to put the previous and next object. The seventh ROI is the rest.

In general, the reasoning needs to be clearer. On lines 164-168: “The presence of fixations on the next target object even before the end of the current action execution shows that the target for the next action is queued in shortly before ending the current action.” This is probably true, but cannot be seen from these data because this is the median end of action execution. You would need to synchronize on the end to really make claims about this. “Moreover, the proportion of fixations on the ’other’ ROI although lower than on the current target shelf, indicates a search process for task-relevant objects/shelves.” Again, maybe true (I am not sure what lower on the shelf is referring to), but impossible to judge from the bundled ‘other’ category. “In sum, in the action execution phase, the results show two primary functions of gaze, first to assist in the completion of the current action and second to locate or search the objects for the next action.” This is certainly true, but you do not need any data for this. Such empty claims, throughout the paper (and the paper is full of them), make it impossible to see when a real finding is being reported. For instance, on the next page all sorts of trivial claims about fixations for planning are made, rather than testing some clear hypothesis about when such fixations take place, or even exploring when they take place. That any difference you see before grasping the next object might contribute to planning is obvious. What are the authors trying to explain?

The summary (from line 222) is also strange: “First, despite large behavioral differences between the two task types, the eye movements do not show corresponding large deviations”. I see no justification for considering the behavioral differences to be large and those on eye movement not to be large, and certainly no justification to directly compare them in this manner, because the behavioral differences are mainly between the movements while the gaze is mainly analyzed within movements. The subsequent claims relating “to relevant cognitive schemas” and “queuing objects into the working memory for subsequent actions” and schemas not corresponding “to broader planning strategies for task completion” are so vague as to be totally meaningless in my opinion. Sentences such as “While look-ahead fixations occurred predominantly before the initiation of the immediate action, the checking fixations occurred in parallel with the action execution” (line 384) are also completely meaningless because they are true by definition. I think the authors need to critically look through the whole paper and remove all sentences that are not actually relevant or meaningful.

Additional details:

Why ignore all fixations with a duration greater than 3.5 times the median absolute deviation of the population fixation duration (line 501)? Does this mean you are removing all instances in which the participant really has to attend to the object in question? Simply removing such fixations can clearly bias the outcome. What happens with the missing time in the analysis?

Why compare the number of displacements with 5000 random possibilities, and not simply with the actual configurations? The authors could identify exactly which trials had additional movements. Were the too easy options removed from the simulations as well (as they were in the data for some reason)? If not, can that explain why the number of movements was lower in the simulations?

Why are the whiskers in Figure 2 1.5 times the IQR and not the actual 95% confidence interval or something like that?

The sentence on line 257 nicely exemplifies what is wrong with the statistics in this study: “In order to statistically analyze the differences in mean relative net transitions in a trial we used a linear mixed effects model”. Such statistics are designed for evaluating a hypothesis, not for exploring differences. The authors are aware of the statistical problems, and conduct corrections for multiple testing to avoid the most obvious ones, but it is not at all clear to me why one would conduct such tests at all if the goal is to explore where the differences might be. Altogether, an emphasis is placed on irrelevant differences, just because they are ‘significant’, whereas the main finding is that they look very much the same. I think the authors need to not just dump all the possible analyses on the reader, but to explain why certain analyses are informative and present those in more detail so that it is evident how the outcome should be interpreted. There is no point dwelling on whether differences at some time interval are significant if you do not even know whether it is correct to match those time intervals (sometimes time itself is considered, and sometimes relative time during an epoch; if the timing is not the same between conditions there must be a difference in one of the two measures). Indeed, without any clear hypothesis the interpretation is meaninglessly vague: lending evidence to “differential gaze guidance behavior” (line 279). Mentioning completely unrealistic options in the argumentation also does not help: e.g. “if fixations are made haphazardly” on line 297 or the fact that the occurrence of first fixations on various items was different than that for the target object on line 339.

Why bother testing the difference between the time taken to perform the tasks with a t-test? Is it relevant whether the difference is statistically significant? My feeling is that it is obvious that there will be a difference, and the question is whether the difference is large enough for one to expect to see differences in gaze related to planning. Or do the authors really consider this to be a hypothesis that needs testing? I will stop with this example because it nicely exemplifies the need to rethink the reasoning behind the experiment, or at least to present such reasoning in a clearer and more concise manner.

Reviewer #2: This manuscript presents an experiment concerning the distribution and timing of eye fixations during a new physical manipulation task. The study is well framed in the literature and provides confirmation of previous findings but also a more detailed picture of how planning and eye-hand coordination evolve during reaching, grasping and placing of objects. The methods used to verify the hypotheses appear adequate and valid and the drawn conclusions warranted. I mostly have some understanding remarks and some interpretation comments, that if discussed, might help the reader get a clearer picture.

1. I assume this is an editorial choice, rather than the authors'choice, but I found a bit annoying that methods were relegated to the end of the manuscript. The transition from the introduction to the results is quite abrupt and I had to retrieve in the last section what participants had to do to have a rough understanding of what the results were about (and also to understand most of Fig. 1). I found the description at the beginning of sec. 2 a bit short and I was left wondering how exactly participants were supposed to execute the tasks. Also, to illustrate the division into planning and execution epochs, Fig. 1D might be referred here.

In section 2.2, the 7 ROIs are suddenly mentioned as a known fact, but these were never introduced before. If no short recall can be given here at least section 4.5.2 should be referred and the legend of Fig. 3.

2. Although I understand the rationale of dividing each "move" in a pre- and post-grasp phase, I wouldn't define them as action planning and action execution epochs. Indeed, in such a task it is somewhat difficult to exactly pinpoint when decision-making and planning take place. The [-2, 2] seconds interval is reasonable but it seems arbitrary and prior to a grasp a reaching movement is issued which is already an action execution. At the same time, after grasp and during placing also planning is at work, as also shown by fixations on the next target. Hence, the temporal distinction between action execution and planning is in my view less strict than what conventionally defined here. It would be actually really interesting to investigate whether such processes (action planning and action control) could be pinpointed on the basis of the oculomotor behavior. Looking at Fig. 4 (and considering Fig.6D) it seems indeed that the different modes in the planning epochs signal different processes at play. While part of the initial fixations are still on the previous target/shelf, probably making sure the object was placed in a stable way, another large portion is on the other region, signaling a searching and decision making behavior between different possible targets, before converging on the actual current target.

3. Following the last comment, it could be helpful to differentiate also the other ROI between other targets and other shelves. This would give some indication whether for example other targets are preferentially looked at prio to the grasp (i.e., deciding which should be the current target) and whether other shelves are considered after the grasp.

4. I do not understand why broader planning strategies were expected, or a higher number of look ahead fixations, but perhaps I misundertood the reasoning. As I gather, and as partly discussed in the conclusion, this task was a) self-paced; 2) did not prescribe a strict sequence of steps. In both the easy and hard case, at any time step a target must be chosen complying with the current task, independently of future targets (e.g., if one is filling a row with red targets, it just needs to find the next red target). At most, in the hard trials, the participant needs to check (with checking fixations) that the same color or shape is not already present in the considered row/column, but they don't need to plan two or more steps ahead, rather indeed they just use the world as an outside memory.

5. Similarly, I suspect there is some possibly culture-based implicit strategy behind the detected spatial heuristics. Again, participants were not constrained in the order in which they sorted the targets but since in western countries writing goes left to right and top to bottom, participants sorting stuff might be biased to fill the shelves this way. This would also explain why usually the most right columns and the most bottom row were left out (since there were 16 objects and a 5x5 grid). This is just an intuition, I do not have pointers to the literature in this sense. Since the task is self-paced, no optimality is enforced or rewarded (e.g., to minimize the displacements) and such a left to right/top to bottom strategy would simplify the step to step planning by offloading the decision to the spatial structure of the grid and the usual scanning strategy.

6. The results section is interesting but really lengthy, perhaps less relevant results could be condensed.

Minor comments:

-line 19-20 it should read "Studying eye movements in moving subjects..."

-line 23: "Kollmorgen et al (2010) demonstrated THAT...."

-line 40 would be best reformulated like "Moreover, Land and Hayhoe(2001) have broadly categorized fixations while performing habitual tasks into four functional groups."

-lines 86 + 88: I'd say realistic environments rather than natural...it's still a human-made environment

-how are the maps in Fig 2 C and D computed? its the overall frequency of displacements across participants?

- line 164: as pointed out above, before and more than an increase on the next target object there's an increase in the other ROI fixations, again this might signal a look-ahead behavior in selecting the next target before ending the grasp (but yet it would need differentaiting between other target rois and other shelves rois)

-line 228-229: again, I'd say that both task types do not require to plan further ahead than the next step

-line 265-266: this could be evidence that indeed the distinction between planning and execution epochs in an artificial one and the two processes are more intertwined

-line 356: it should read "during action planning" instead of while

Reviewer #3: This paper is an interesting piece of work that highlights with more systematic experimentation some of the early gaze control work by Hayhoe/Ballard in the 90s-00s, using a nicely-structured virtual reality task. The fact that the task is not overlearned is I agree an important point. I would have maybe also liked a small paragraph or two discussing the translatability of these findings in a VR setup to a similar experimental setup in the real world.

- The opening sentence I think maybe needs a bit of revision, I'm not sure what 'in the pragmatic turn' means in this context. I very much agree with the idea about moving from embodiment to cognition, the arguments for are quite strong.

- Only lists refresh rate of the HMD, not of the Tobii eye-trackers in the paper as far as I can see (the frequency from the manufacturer specs appears to be 120Hz, with 0.5-1.1 degree accuracy)

For Figure 2, it looks like the plots for 2B have changed slightly between the preprint and this submitted version, in terms of the average values and error bars - is this due to more trials being performed, or a re-analysis of the existing data?

For Figure 3, it's not very clear what the main scientific point is with respect to the differences between the horizontal traces - what point is the reader meant to take away from these in particular? It would be good to have a little box highlighting the region that is 'expanded' in the latter section of Figure 3, it wasn't very clear on initial viewing. Why also was 2 seconds the size of the window selected, was this arbitrary or based off a precedent from other work?

For Figure 4, 'Region of Interests' should probably read 'Regions of Interest'. Line 294 should probably read 'median time to first fixation, rather than 'median time to first fixate on', but on the whole otherwise it's a good figure.

Overall, in my opinion this paper would be a good fit for PLOS Computational Biology, and overall I consider it well-formed and a good study that begins to answer some outstanding questions in our field.

**Have the authors made all data and (if applicable) computational code underlying the findings in their manuscript fully available?**

Reviewer #1: Yes

Reviewer #2: Yes

Reviewer #3: Yes

PLOS authors have the option to publish the peer review history of their article (what does this mean?). If published, this will include your full peer review and any attached files.

Reviewer #1: No

Reviewer #2: No

Reviewer #3: No
---

## [Decision Letter · Decision Letter 1]

30 Apr 2024

Dear Ms Keshava,

Thank you very much for submitting your manuscript "Just-in-time: Gaze Guidance in Natural Behavior" for consideration at PLOS Computational Biology. As with all papers reviewed by the journal, your manuscript was reviewed by members of the editorial board and by several independent reviewers. The reviewers appreciated the attention to an important topic. Based on the reviews, we are likely to accept this manuscript for publication, providing that you modify the manuscript according to the review recommendations.

Sincerely,

Aldo A Faisal

Academic Editor

PLOS Computational Biology

Wolfgang Einhäuser

Section Editor

PLOS Computational Biology

Reviewer's Responses to Questions

**Comments to the Authors:**

Reviewer #1: The revision has certainly improved the paper considerably, but I think the presentation could be even better in a couple of places. In my opinion, rather trivial findings are sometimes presented in too much detail, while important issues are hardly discussed. An example of the latter is that I wonder whether the conclusion that the study demonstrates that gaze is controlled in a just-in-time manner (lines 534-535) is justified. The term just-in-time suggests that gaze is directed to objects just before the planned arm movement starts, but actually the converse could also be true, that the hand only starts moving when gaze has located the target. Indeed, in the abstract the authors seem to be proposing the latter interpretation. I think the authors should somewhere specify precisely what they mean, because now ‘just-in-time’ seems to be used in both senses (which is also fine, I guess, but then it should be specified that no distinction is made between the two).

Another issue is the comparison with optimal behaviour. Since this appears to be an important aspect of the study, I wonder whether it would not be worthwhile expanding the analysis a bit by also considering a comparison with optimal behaviour under the constraints that there should be no empty row or column between filled ones, and/or that the final configuration should be in the top left (so leaving the rightmost column and bottom row empty). Is this enough to explain performance not being optimal? I think this would be interesting to know, irrespective of the outcome. It would show how good an overview the participants have of the whole situation. Below I will propose yet another option.

I really like Figure 3, but I feel that the emphasis in the text is on the differences between hard and easy trials, but the most striking aspect to me is the similarity. The differences are easy to explain, and yet they are presented as interesting findings. We know from Figure 2B1 that preparation takes 2s in easy trials but 4s in hard trials. So if participants are looking at the current object until and just after drop-off time (right side of Figure 3) irrespective of the trial type, it is evident that this will be seen as more gaze at the previous target object about 2s before (the next) grasp onset on easy trials (which on average is the same moment) but not on hard trials (where the same moment is much earlier). Similarly, looking more at other objects for hard trials is logical because the next target you look at is more likely to be ‘wrong’ (and therefore ‘other’) if there are more constraints. I think all this is interesting, but it would be more useful to explain some of these things than to test (and reject) the completely unrealistic hypothesis that there would be no difference.

Minor issues

The statement “The average fixations aligned to the action onset revealed gaze guidance was driven just in time” on line 451-452 ignores the possibility that the action starts as soon as the target is found. Thus, gaze is not just in time but the action starts when the target is found. 

Is the fact that a depth-first search model was used to find the optimal behaviour relevant? Was the search done to a limited depth? Otherwise I would remove the detail or explain why this particular search method was best here. Providing irrelevant details distracts the reader. I started wondering why this particular method was used, when probably it was just an arbitrary choice because it should not matter what kind of search is used.

The authors did not understand my concern with removing ‘outliers’. My concern is not about the method, but about whether one should really consider them to be outliers. By doing so the authors are assuming that participants should always be doing the same thing, with random variability, but there may be a reason to sometimes fixate a certain object for a long time. It was just a thought. The authors do not need to do anything about this if they do not want to, but they should realise the assumption that they are making.

The task (line 114) does not make sense before the reader knows what the objects are, so that needs to be added at the beginning of the results section. Only when you know that there are 4 different shapes, each in 4 different colours, does the HARD instruction make sense.

On line 124 I could not figure out what ‘gaze position and direction data’ means. From the methods it becomes apparent that gaze direction was used to determine angular velocity. Maybe remove or explain ‘position’.

I wonder about the analyses that include index (Figure 2B2). We are shown a fit line, without data points. In this case, I think showing the data points might be informative. It makes sense for the preparation to take more time for hard trials, but I would have expected a different slope. The last point(s) could have been fast, because there are few options left so it should be easier to decide. The fact that the first movement was fast corresponds well with the fact that participants did not optimise their choice of row and column but simply used the top left part of the shelves. If participants do not minimise the number of movements they might just first randomly place appropriate objects on the first row, ignoring conflicts with objects already present on other rows, in which case it makes sense for the first row to be fastest. I think adding the values as faint dots (as in B1) or at least adding the means for each grasp index might be informative as to such strategies. The latter strategy could even also be simulated to see whether the actual performance is consistent with this. I think analysing the data in more detail might be much more informative than the current statistical approach. It is evident from the figure that participants are not optimal, but testing the null hypothesis is very tricky, because they cannot be better than optimal so any test based on normal distributions cannot really be valid. I think the results are very clear (e.g. figure 2C) but the statistical tests leave me with the impression that I am being fooled. And this while the data themselves are so interesting, and probably could even reveal more about how the task is done.

I think the authors could specify in lines 208-210 how specifically subjects “offset their cognitive load” and what kind of “sub-optimal strategy” they could have adopted. I have made some suggestions above (pick rows and columns in advance; ignore conflicts with items on future rows). I think a combination of looking at the preparation times and at the placing orders might actually reveal how participants do this.

An issue that was not quite clear to me was whether the subjects were instructed to make as few moves as possible or to be as fast as possible. Maybe this should explicitly be mentioned, even if no such instruction was given, because ‘optimality’ is defined by the authors in terms of minimising the number of moves, but maybe the participant can be faster if he or she does not worry about making additional moves (saving preparation time; maybe this is what the authors have in mind when they mention reducing cognitive load, although there may be other reasons to reduce cognitive load than reducing the required time).

In my opinion the analysis of the gaze transition behaviour is superfluous and could be removed. It is a clever analysis, but the outcome is trivial. Of course the transitions will be more variable for the hard task. This is related to looking more at other objects. At least this should be discussed rather than this analysis being presented as something independent. But doing an analysis and then discussing that the results are trivial seems like a waste of time and space to me.

The most obviously superfluous analysis, in my opinion, is that described in section 2.4. Figure 5 shows the order of fixations in a clear manner. It took me a long time to figure out what the statistical analysis means and shows, just to discover that the conclusion is best supported by figure 5 rather than by all the statistics and numbers given in the text. Moreover, the conclusion on line 431 is trivial. It could not have been the other way around.

The first part of section 5.4.1 seems to me to be a detail of the way in which the data were provided by the eye tracker. Apparently the eye tracker gives the direction as a unit vector rather than as two angles or as the position on the screens, but why bother the reader with this? It just makes the paper more complicated.

The statement on line 637 that the ‘other’ items were ‘irrelevant to the action sequence’ is strange. To perform optimally you may need to look at them to determine that they are not the correct item. The authors appear to only consider gaze for manipulating objects, not for finding them.

On line 650 the authors mention ‘novelty’, but what this means is only explained later.

There is a typo on line 116: unique shape (not shapes)

Reviewer #2: I thank you the authors for the substantial work in addressing the reviewers' comments. I think the manuscript has improved.

I just have minor comments:

- in many places 'proactive actions' are mentioned but I'm not sure what is meant by that: proactive to what? The manual actions do execute the plan. The gaze control, possibly, is proactive, retrieving information in advance and allowing actions to be executed.

- I would reccomend a thorough proof-reading because of some typos and awkward formulations, e.g.:

pag 3, line 116: it should read unique shape, not shapes

pag 4, fig, 1.B caption: "The objects were randomly presented

on a 5x5 shelf at the beginning of each trial and were cued to sort objects by shape and color." I suspect what is meant is that participants were cued to sort objects?

pag 5, line 126:" we classified fixations to gaze samples...", perhaps you classified as fixation a cluster of gaze samples with duration larger than 100 ms?

Reviewer #4: The authors conducted a VR study in which they looked at gaze patterns depending on task complexity. The paradigm used is new to me and sounds like a promising approach to investigate the question. In my opinion, there is a successful descriptive analysis of the data. The study is also a good example of how we can use VR studies to investigate eye movements (or actions in general).

General Comment:

The introduction initially gave me the impression that a new model was being presented. It is said that the Hayhoe/Ballard model (which is also presented in Figure 1A) is being generalized. For me, this generalization or extension does not really exist. I cannot find an extended scheme or a new model. Rather, there are descriptive results that are analyzed and classified. Maybe it would help to create another illustration like 1A, where exactly the innovation lies? As I said in my initial comment, I think the study is definitely suitable for publication, but the scope should be more clearly defined. It is a new paradigm for investigating the planning of gaze sequences, but in my view it is not a new or generalised framework for describing them. Where is the classification of existing computational models of eye movements in planning? (e.g. https://www.nature.com/articles/s41598-018-37536-0). In my opinion (if the publication is not supposed to change much, such as a development of a computational model), there should be a clearer delimitation of the contribution, namely that it is a new paradigm and a first experimental investigation and not a new framework.

Additional minor points:

- The linked repository should be updated. The status is still from before the revisions.

- Figure 1C could be moved to the methods section. The first figure should give a general overview of the paper, the study and the motivation. For me, such technical details have no place here. Maybe throw them out and move Panel B to this position?

- Table 1 is incorrectly placed. It is on page 5 but is referenced on page 7. In between is Figure 2.

- Why is Figure 2 not arranged regularly? It is visually very confusing that it does not go from left to right and from top to bottom.

- The conclusions regarding the analyses in Figure 2D and E are not far-reaching enough for me. A clear effect is observed and reported here. However, the subsequent classification is rather brief. It is still unclear to me why there is this spatial bias here and, above all, why it looks the way it does. You should go into this in more detail.

- Where can I find details on the depth search model? I couldn't find any in the paper or in the repository. Also, would it be an idea to compare whether humans also perform depth search (or breadth-first search)?

- Figure 3 overwhelms me visually. I can't extract any information from it. The line plot is already difficult, but together with the horizontal bars I can't see through it at all. Can you perhaps break this down a bit more?

- Line 517 and following: I wouldn't necessarily classify VR as an absolute new paradigm. The paragraph almost reads as if there is hardly any existing work on it and that it has only been in use for 1.2 years. I would tone it down a bit and add more relevant literature.

**Have the authors made all data and (if applicable) computational code underlying the findings in their manuscript fully available?**

Reviewer #1: Yes

Reviewer #2: None

Reviewer #4: **No: **As noted in my review, the repository is not up to date (but presumably up to date from the initial submission). Also, I could not find any details on the depth-first search model.

PLOS authors have the option to publish the peer review history of their article (what does this mean?). If published, this will include your full peer review and any attached files.

Reviewer #1: No

Reviewer #2: No

Reviewer #4: No

Figure Files:

Data Requirements:

Reproducibility:

References:

---

## [Decision Letter · Decision Letter 2]

1 Oct 2024

Dear Ms Keshava,

We are pleased to inform you that your manuscript 'Just-in-time: Gaze Guidance in Natural Behavior' has been provisionally accepted for publication in PLOS Computational Biology.

Best regards,

Aldo A Faisal

Academic Editor

PLOS Computational Biology

Wolfgang Einhäuser

Academic Editor

PLOS Computational Biology

Reviewer's Responses to Questions

**Comments to the Authors:**

Reviewer #1: I like this version of the paper. I think participants may have behaved differently if the instructions were slightly different. If they knew that you woud be evaluating their performance in terms of the number of moves, they might have spent more time searching and not have displayed the tendency to move objects to the upper left. Looking at the influence of the instruction might be a nice follow up experiment. In that case some of the differences between the two tasks might become even bigger. But that is for the future.

Reviewer #2: Thank you for undertaking this revision, I do not have any further comments

**Have the authors made all data and (if applicable) computational code underlying the findings in their manuscript fully available?**

Reviewer #1: Yes

Reviewer #2: Yes

PLOS authors have the option to publish the peer review history of their article (what does this mean?). If published, this will include your full peer review and any attached files.

Reviewer #1: No

Reviewer #2: No

---

## [Editor Report · Acceptance letter]

16 Oct 2024

PCOMPBIOL-D-22-00214R2 

Just-in-time: Gaze Guidance in Natural Behavior

Dear Dr Keshava,

I am pleased to inform you that your manuscript has been formally accepted for publication in PLOS Computational Biology. Your manuscript is now with our production department and you will be notified of the publication date in due course.

With kind regards,

Zsuzsanna Gémesi
